# Pushing the limits of self-supervised ResNets: Can we outperform supervised learning without labels on ImageNet?

## Abstract

Despite recent progress made by self-supervised methods in representation learning with residual networks, they still underperform supervised learning on the ImageNet classification benchmark, limiting their applicability in performance-critical settings. Building on prior theoretical insights from RELIC [Mitrovic et al., 2021], we include additional inductive biases into self-supervised learning. We propose a new self-supervised representation learning method, RELICv2, which combines an explicit invariance loss with a contrastive objective over a varied set of appropriately constructed data views to avoid learning spurious correlations and obtain more informative representations. RELICv2 achieves 77.1% top-1 classification accuracy on ImageNet using linear evaluation with a ResNet50 architecture and 80.6% with larger ResNet models, outperforming previous state-of-the-art self-supervised approaches by a wide margin. Most notably, RELICv2 is the first unsupervised representation learning method to consistently outperform the supervised baseline in a like-for-like comparison over a range of ResNet architectures. Finally, we show that despite using ResNet encoders, RELICv2 is comparable to state-of-the-art self-supervised vision transformers.

## 1 Introduction

Large-scale *foundation models* [Bommasani et al., 2021]—in particular for language [Devlin et al., 2018; Brown et al., 2020] and multimodal domains [Radford et al., 2021]—are an important recent development in representation learning. The idea that massive models can be trained without labels in an unsupervised (or self-supervised) manner and be readily adapted, in a few- or zero-shot setting, to perform well on a variety of downstream tasks is important for many problem areas for which labeled data is expensive or impractical to obtain. *Contrastive* objectives have emerged as a successful strategy for representation learning [Chen et al., 2020a; He et al., 2019]. However, downstream utility[1] of these representations has until now never exceeded the performance of supervised training of the same architecture, thus limiting their usefulness.

In this work, we tackle the question "Can we outperform supervised learning without labels on ImageNet?". As such, our focus is on learning good representations for high-level vision tasks such as image classification. In supervised learning we have access to label information which provides a signal of what features are relevant for classification. In unsupervised learning there is no such signal and we have strictly less information available from which to learn compared to supervised learning. Thus, the challenge of outperforming supervised learning without labels might seem impossible. Comparing supervised and contrastive objectives we see that they are two very different approaches to learning representations that yield significantly different representations. While supervised approaches use labels as targets in within a cross-entropy objective, contrastive methods rely on comparing against similar and dissimilar datapoints. Thus, supervised representations end up encoding a small set of highly informative features for downstream performance, while contrastive representations encode many more features with some of these features not related to downstream performance. This intuition is also supported by the observation that when visualizing the encoded

---

[1] Downstream utility is commonly measured by how well a method performs under the standard linear evaluation protocol on ImageNet; see section 3.

information of contrastive representations through reconstruction, they are found to retain more detailed information of the original image, such as background and style, than supervised representations [Bordes et al., 2021].

Based on this, we hypothesize that one of the key reasons for the current subpar performance of contrastive (and thus self-supervised) representations, is the presence of features which are not directly related to downstream tasks, i.e. so-called *spurious features*. In general, basing representations on spurious features can have negative consequences for the generalization performance of the model and thus avoiding to encode these features is paramount for learning informative representations.

In this paper, we propose to equip self-supervised methods with additional inductive biases to obtain more informative representations and overcome the lack of additional information that supervised methods have access to. We use as our base self-supervised approach the performant RELIC method

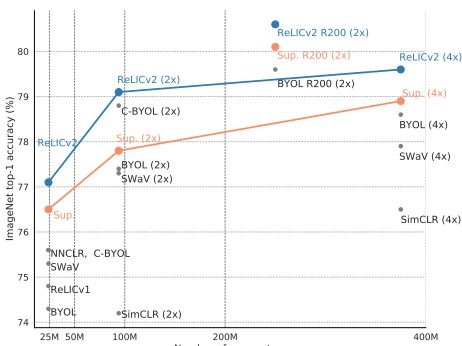

Figure 1: Top-1 linear evaluation accuracy on ImageNet using ResNet50 encoders with $1\times$, $2\times$, $4\times$ width multipliers and a ResNet200 with a $2\times$ width multiplier.

[Mitrovic et al., 2021] which combines a contrastive loss with an invariance loss. We propose to extend RELIC by adding inductive biases that penalize the learning of spurious features such as background and style, e.g. brightness; we denote this method as RELICv2. First, we propose a new fully unsupervised saliency masking pipeline which enables us to separate the image foreground from its background. We include this novel saliency masking approach as part of the data augmentation pipeline. Leveraging the invariance loss, this enables us to learn representations that do not rely on the spurious feature of background. Second, while RELIC operates on just two data views of the same size, we extend RELICv2 to multiple data views of varying sizes. Specifically, we argue for learning from a large number of data views of standard size as well as including a small number of data views that encode only a small part of the image. The intuition for using smaller crops is that this enables learning of more robust representations as not all features of the foreground might be contained in the small crop. Thus, the learned representation is robust against individual object features being absent as it is the case in many real-world settings, e.g. only parts of an object are visible because of occlusion. On the other hand, using multiple large crops enables us to learn representations that are invariant to object style. We extend the contrastive and invariance losses of RELIC to operate over multiple views of varying sizes.

Empirically we show RELICv2 achieves state-of-the-art performance in self-supervised learning on a wide range of ResNet architectures (ResNet50, ResNet 50 2x, ResNet 50 4x, ResNet101, ResNet152, ResNet200, ResNet 200 2x). As shown in figure 1, RELICv2 is the first self-supervised representation learning method that outperforms a standard supervised baseline on linear ImageNet evaluation across a wide range of ResNet architectures. On top-1 classification accuracy on ImageNet, RELICv2 achieves $77.1\%$ with a ResNet50, while with a ResNet200 $2\times$ it achieves $80.6\%$. Furthermore, RELICv2 learns representations that achieve state-of-the-art performance on a wide range of downstream tasks and datasets such as semi-supervised and transfer learning, robustness and out-of-distribution generalisation. Although using ResNets, RELICv2 demonstrates comparable performance to recent vision transformers (figure 6). These strong experimental results across different vision tasks, learning settings and datasets showcase the generality of our proposed representation learning method. Moreover, we believe that the concepts and results developed in this work could have important implications for wider adoption of self-supervised pre-training in a variety of domains as well as the design of objectives for foundational machine learning systems.

**Summary of contributions.** We tackle the question "Can we outperform supervised learning without labels on ImageNet?". We propose to extend the self-supervised method RELIC [Mitrovic et al., 2021] with inductive biases to learn more informative representations. We develop a fully unsupervised saliency masking pipeline and use multiple data views of varying sizes to encode these inductive biases. The resulting method RELICv2 achieves state-of-the-art performance in self-supervised learning across a wide range of ResNet architectures of different depths and width. Furthermore, RELICv2 is the first self-supervised representation learning method that outperforms a standard supervised ResNet50 baseline on linear ImageNet evaluation across $1\times$, $2\times$ and $4\times$ ResNet50 vari-

ants as well as on larger ResNet architectures such as ResNet101, ResNet152 and ResNet200.[2] We also highlight the generality of RELICv2 through state-of-the-art self-supervised performance on transfer learning, semi-supervised learning, and robustness and out-of-distribution generalization.

## 2 METHOD

### 2.1 BACKGROUND

Representation Learning via Invariant Causal Mechanisms (RELIC) [Mitrovic et al., 2021] learns representations by comparing two differently augmented data views through two mechanisms – instance classification and invariant prediction. As is common in prior work, RELIC tackles the instance classification problem with a contrastive objective, i.e. it learns representations by maximizing the similarity between two differently augmented views of the same datapoint (positives), while minimizing the similarity of that datapoint with other datapoints (negatives). In addition to that, RELIC introduces an invariance objective which ensures that the distribution of similarities of a datapoint is invariant across augmentations of that datapoint.

Given a randomly sampled batch of datapoints $\{x_i\}_{i=1}^N$ with $N$ the batch size, RELIC learns an encoder $f$ that outputs the representation $z$, i.e. $z_i = f(x_i)$. Following [Chen et al., 2020a; Grill et al., 2020], RELIC creates two views of the data by applying two distinct augmentations randomly sampled from the data augmentation pipeline proposed in [Chen et al., 2020a] , i.e. $t, t' \sim \mathcal{T}$; this yields two augmented batches $\{x_i^t\}_{i=1}^N$ and $\{x_i^{t'}\}_{i=1}^N$. To solve the instance classification problem, RELIC maximizes the following probability

$$p(x_i^t; x_i^{t'}) = \frac{e^{\phi_\tau\left(x_i^t, x_i^{t'}\right)}}{e^{\phi_\tau\left(x_i^t, x_i^{t'}\right)} + \sum_{x_j^{t'} \in \mathcal{N}(x_i)} e^{\phi_\tau\left(x_i^t, x_j^{t'}\right)}} \tag{1}$$

where $\phi_\tau(x_i, x_j) = \langle h(f(x_i)), q(g(x_j)) \rangle / \tau$ measures the similarity between representations with $\tau$ the temperature parameter. RELIC adopts the *target* network setting of [Grill et al., 2020] such that $f$ and $g$ have the same architecture, but the weights of $g$ are an exponential moving average of the weights of $f$; also, $h$ and $q$ are multi-layer perceptrons with $h$ playing the role of the composition of the projector and predictor from [Grill et al., 2020] and $q$ being the exponential moving average of the projector network. $\mathcal{N}(x_i)$ represents the set of *negatives*; we uniformly randomly sample a small number of points from the batch to serve as negatives following [Mitrovic et al., 2020].

In order to optimize for invariant prediction, RELIC introduces an *invariance loss* defined as the Kullback-Leibler divergence between the likelihood of the two augmented data views as

$$D_{\mathrm{KL}}(p(x_i^t)|p(x_i^{t'})) = \mathrm{sg}\left[\mathbb{E}_{p(x_i^t;x_i^{t'})} \log p(x_i^t; x_i^{t'})\right] - \mathbb{E}_{p(x_i^t;x_i^{t'})} \log p(x_i^{t'}; x_i^t). \tag{2}$$

The invariance loss enforces that the similarity of $f(x_i^t)$ and $f(x_i^{t'})$ *relative* to the positives and negatives is the same; the stop gradient operator, $\mathrm{sg}[\cdot]$, does not affect the computation of the KL-divergence but avoids degenerate solutions during optimization [Xie et al., 2019]. Taken together, RELIC learns representations by optimizing the following loss

$$\mathcal{L} = \sum_{t,t'\sim\mathcal{T}} \sum_{i=1}^N -\log p(x_i^t; x_i^{t'}) + \beta D_{\mathrm{KL}}(p(x_i^t)|p(x_i^{t'})) \tag{3}$$

with $\beta$ a scalar weighting the relative importance of the contrastive and invariance.

### 2.2 RELICv2

We extend Representation Learning via Invariant Causal Mechanisms (RELIC) [Mitrovic et al., 2021] to include additional inductive biases that prevent the learning of spurious features which are hurting the learning of informative representations. We encode these inductive biases through a novel fully unsupervised saliency masking pipeline as well as scaling up RELIC to accommodate multiple data views of varying sizes; we call this method RELICv2 which we highlight in figure 2.

---

[2]Concurrent work in [Lee et al., 2021] outperforms the same standard supervised baseline only on a ResNet50 $2\times$ encoder.

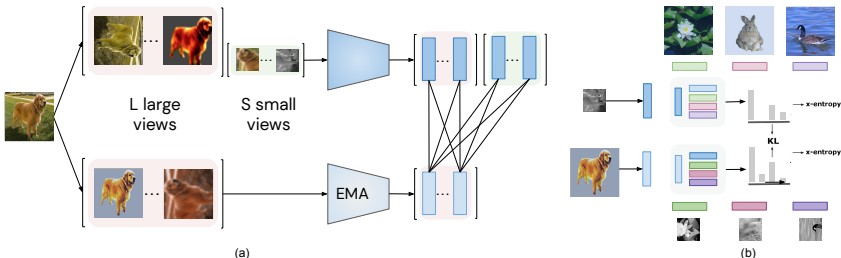

(a)                                                    (b)

Figure 2: (a) RELICv2 uses saliency masking as part of the data augmentations pipeline and views of various sizes to learn representations that are invariant to spurious correlations. Note that the $L$ (differently augmented) large views are passed through both the encoder and target network, while the small $S$ views are only passed through the encoder. The learning objective is computed by comparing each of the large and small view passed through the encoder with each large view passed through the target. (b) Learning objective used for each comparison that combines the contrastive (instance discrimination) loss, i.e the cross-entropy (x-entropy) loss computing the similarity scores and the invariance loss, i.e. the Kullback-Leibler (KL) divergence between the similarity scores.

**Saliency masking.** To localize the semantically relevant parts of the image, we propose to use saliency masking. In order to separate foreground from background, we develop a fully unsupervised saliency masking pipeline that does not use any additional data. We learn a saliency masking network without access to any labels or any additional data outside of the ImageNet training dataset which we also use for pre-training the representation. We start by using a small subset of ImageNet (2500 randomly selected images) and compute initial estimates of saliency masks using a number of handcrafted methods. Specifically, we use Robust Background Detection [Zhu et al., 2014], Manifold Ranking [Yang et al., 2013], Dense and Sparse Reconstruction [Li et al., 2013] and Markov Chain [Jiang et al., 2013]. Next, we use a ResNet50 2x network pretrained on ImageNet in a self-supervised way as the initial saliency detection network. We adopt the self-supervised refinement mechanism from DeepUSPS[3] [Nguyen et al., 2019] and incrementally refine the handcrafted saliency masks using a self-supervised objective. Subsequently, the saliency detection network is trained by fusing the refined handcrafted saliency masks. The trained saliency detection network is used for computing the saliency masks in our pipeline. During RELICv2 pre-training, we randomly apply the saliency mask to input images with a small probability $p_m$. By learning from images where the background has been removed RELICv2 is learns representations that focus on foreground features and are robust to background changes as well as any spurious features that might be found in the background. For more details see appendix.

**Views of varying sizes.** We propose to enforce invariance over multiple randomly augmented data views of varying sizes to learn more informative representations. We propose to use a large number of views encoding the whole randomly augmented image as well as a small number of smaller views which contain only a portion of the randomly augmented image.[4] As the number of views of the whole image increases so does the variation of object styles seen during training as each view represents a random augmentation manipulating the style of the image. Thus, explicitly enforcing invariance over an increasing set of object styles enables us to learn representations which are increasingly invariant to the spurious changes in object style. Incorporating small views which are random crops of the original image into the learning serves two purposes. First, as these views represent a small part of the original image, it is likely that some parts of the objects of interest might be occluded. Thus, as we learn representations through invariance, using these small views enables us to learn representations which are more robust to object occlusions, a common issue in real-world data. Note that it is important to use only a small number of small views (in comparison to the number of large views) as using too many small views can result in representations that fail to encode important features as different features might be occluded in different small views and we are learning representations by enforcing invariance.[5] Second, we hypothesize that small crops

---

[3]Note that DeepUSPS cannot be directly applied in our setting as it relies on labelled information and additional datasets (CityScapes) [Cordts et al., 2016].

[4]Most other methods use only 2 data views of the whole image.

[5]For this purpose, we use only 2 small views in our experiments and provide an ablation analysis on the different number of large and small views.

play a synergistic role to saliency masking as taking a small crop of the image is likely to remove potentially large parts of the background; see analysis section for some experimental validation.[6]

**Objective.** Let $x_i^t$ be a large size view and $\tilde{x}_i^t$ a small size view under augmentation $t \sim \mathcal{T}_{\text{sal}}$, respectively. RELICv2 optimizes a combination of the contrastive and invariance losses across differently augmented large and small views. In particular, RELICv2 optimizes the following objective:

$$\mathcal{L} = \sum_{i=1}^{N} \sum_{1 \leq l_1 \leq L} \left( \sum_{1 \leq l_2 \leq L} \left( -\log p(x_i^{t_{l_2}}; x_i^{t_{l_1}}) + \beta D_{\text{KL}}(p(x_i^{t_{l_2}})|p(x_i^{t_{l_1}})) \right) + \tag{4} \right.$$
$$\left. \sum_{1 \leq s \leq S} \left( -\log p(\tilde{x}_i^{t_s}; x_i^{t_{l_1}}) + \beta D_{\text{KL}}(p(\tilde{x}_i^{t_s})p(x_i^{t_{l_1}})) \right) \right)$$

with $t_{l.} \sim \mathcal{T}_{\text{sal}}$ and $t_s \sim \mathcal{T}$ randomly selected data augmentations, and $L$ and $S$ the number of large and small crops, respectively. Note that we leverage the differently sized data views in different ways for learning representations. We use the large views both for updating the encoder $f$ as well as for computing learning targets through the target network $g$, i.e. $x_i^{t_{l.}}$ appears on both sides of $p$. On the other hand, we only use the small views for updating the encoder $f$ and not as learning targets, i.e. $\tilde{x}_i^s$ appears only on the left hand side of $p$, c.f. equation 1. We do not use small views as learning targets as potentially informative parts of the image might be occluded and as such the corresponding features removed from the representation. For the precise architectural and implementation details, as well as a pseudo-code for RELICv2 see the appendix.

## 3 OUTPERFORMING SUPERVISED LEARNING ON IMAGENET

We use RELICv2 to pretrain representations on the training set of the ImageNet ILSVRC-2012 dataset [Russakovsky et al., 2015] without using labels. As our supervised baseline we use the supervised ResNet50 model used through the representation learning literature (c.f. [Chen et al., 2020a; Grill et al., 2020; Caron et al., 2020; Dwibedi et al., 2021]). This model is trained with a cross-entropy loss, a cosine learning rate schedule and full access to labels using the same set of data augmentations as proposed by [Chen et al., 2020a]. While the recent work of [Wightman et al., 2021] proposes a series of elaborate optimization and data augmentation tricks to improve the performance of a supervised ResNet50 on ImageNet, we leave the application of these heuristics for future work. Instead, in this work, we focus on a fair like-for-like comparison between self-supervised methods and supervised learning [7].

Table 1: Top-1 accuracy (in %) under linear evaluation on the ImageNet test set for a ResNet50 encoder set for different representation learning methods.

| Method | Top-1 |
|---|---|
| Supervised [Chen et al., 2020a] | 76.5 |
| SimCLR [Chen et al., 2020a] | 69.3 |
| MoCo v2 [Chen et al., 2020b] | 71.1 |
| InfoMin Aug. [Tian et al., 2020] | 73.0 |
| BYOL [Grill et al., 2020] | 74.3 |
| RELIC [Mitrovic et al., 2021] | 74.8 |
| SwAV [Caron et al., 2020] | 75.3 |
| NNCLR [Dwibedi et al., 2021] | 75.6 |
| C-BYOL [Lee et al., 2021] | 75.6 |
| RELICv2 (ours) | **77.1** |

We evaluate RELICv2's representations by training a linear classifier on top of the frozen representation according to the procedure described in [Chen et al., 2020a; Grill et al., 2020; Dwibedi et al., 2021] and the appendix. From table 1, for the ResNet50 encoder, we see that RELICv2 outperforms all previous state-of-the-art self-supervised approaches by a significant margin in terms of top-1 accuracy on the ImageNet test set. Remarkably, RELICv2 outperforms the standard supervised baseline in terms of top-1 accuracy despite using no label information to pretrain the representation.

In table 2, we also note that RELICv2 outperforms the supervised baseline under a varied set of ResNet encoders of different sizes, spanning ResNet50, ResNet101, ResNet152 and ResNet200. Refer to the appendix for additional results and more details. Overall, our work can be viewed as a first step in surpassing supervised learning without access to labels across many different ResNet architectures for the image classification task. Furthermore, RELICv2 also performs competitively to the latest vision transformer architectures at similar parameter counts (see figure 6).

---

[6]Note that saliency masking is not perfect at removing the background and struggles especially in settings where there is little color contrast between the background and foreground.

[7]Note that using saliency masking to create background augmentations for training the supervised baseline does not improve performance as highlighted in Ryali et al. [2021b]

Table 2: Top-1 acc. (in %) under linear evaluation on ImageNet for different ResNet architectures.

| Method | ResNet50 2x | ResNet50 4x | ResNet101 | ResNet152 | ResNet200 | ResNet200 2x |
|---|---|---|---|---|---|---|
| Supervised [Chen et al., 2020a; Grill et al., 2020] | 77.8 | 78.9 | 78.0 | 79.1 | 79.3 | 80.1 |
| BYOL [Grill et al., 2020] | 77.4 | 78.6 | 76.4 | 77.3 | 77.8 | 79.6 |
| RELICv2 (ours) | **79.0** | **79.4** | **78.7** | **79.3** | **79.8** | **80.6** |

**Scaling.** Figure 3 shows the ImageNet linear evaluation accuracy obtained by representations learned using RELICv2 as a function of the number of images seen during pre-training using the ImageNet training set. It can be seen that in order to reach 70% accuracy the ResNet50 model requires approximately twice the number of iterations as the ResNet295 model. The ResNet295 has approximately $3.6\times$ the number of parameters as the ResNet50 (87M vs 24M, respectively). This finding is in accordance with other works which show that larger models are more sample efficient (i.e. they require fewer samples to reach a given accuracy) [Zhai et al., 2021].

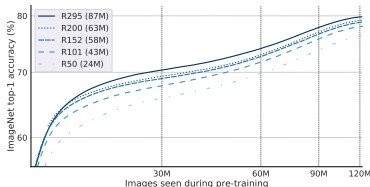

Figure 3: ImageNet accuracy obtained by RELICv2 as a function of number of images seen during pre-training for several of ResNet architectures (Number of model parameters in brackets).

### 3.1 ANALYSIS

**Class concentration.** To quantify the overall structure of the learned latent space, we examine within- and between-class distances. Figure 4 compares the distribution of ratios of between-class and within-class $\ell_2$-distances of the representations of the ImageNet test set learned by RELICv2 against those learned by the supervised baseline.[8]

A larger ratio means that the representation is better concentrated within the corresponding classes as well as better separated between classes and therefore more easily linearly separable (c.f. Fisher's linear discriminants [Friedman et al., 2009]). We see that RELICv2's distribution is shifted to the right, i.e. has a higher ratio compared to the supervised baseline, suggesting that the better representations have been learned. For additional analysis of the distances between learned representations of closely related classes, refer to the appendix.

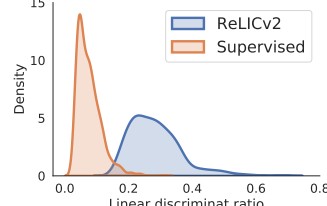

Figure 4: Distribution of the *linear discriminant ratio*: the ratio of between-class distances and within-class distances of embeddings computed on the ImageNet validation set.

**Views of varying sizes.** Most prior work uses 2 views of size $224 \times 224$ to learn representations. RELICv2 proposes instead the use of a large number of views of size $224 \times 224$ combined with a small number of smaller views of size $96 \times 96$. We ablate the use of different numbers of large and small views in RELICv2 using only SimCLR augmentations (i.e. without saliency masking). We report below the top-1 ImageNet test performance (under linear evaluation) of a ResNet50 pretrained for 1000 epoch on ImageNet; $[L, S]$ denotes using L large views and S small views.

| Views | [2, 0] | [2, 2] | [2, 6] | [4, 0] | [4, 2] | [6, 2] | [8, 2] |
|---|---|---|---|---|---|---|---|
| Top-1 | 74.8 | 76.2 | 76.0 | 75.5 | **76.8** | 76.5 | 76.5 |

There seems to be a performance plateau going beyond 6 large views and a slight performance penalty going beyond 4 large views. For small views, we also observe performance penalties going beyond 2 small views, while there is a significant performance boost going from no small views to 2 small views, i.e. $+1.3\%$ in the case of 4 large views. This is almost double the performance improvement one gets from adding 2 large views, i.e the difference between $[2, 0]$ and $[4, 0]$ of $+0.7\%$. This supports our hypothesis that small views significantly contribute to learning more robust representations, but that we only need a small number of them to learn informative representations. Note that our use of large and small views is exactly the opposite as compared to SWaV [Caron et al., 2018] which argue for using smaller views as computationally less expensive alternatives to large views and rely heavily on using a large number of small views; in particular, they argue for using 2 large views and 6 small views.

---

[8]Both RELICv2 and the standard supervised baseline were trained on the ImageNet training set.

**Saliency masking**. To isolate the contribution of saliency masking, we measure the performance gain (in terms of top-1 accuracy under linear evaluation on ImageNet) when applying saliency masking to just 2 large views. This improves performance from $74.8\%$ to $75.3\%$, i.e. a gain of $+0.5\%$ which is a boost comparable to having two additional large views (see above). We also explored using different datasets for pretraining our unsupervised saliency masking pipeline as well as varying the probability with which we apply saliency masking to the inputs and found that RELICv2 is robust to these choices; see the appendix for details.

**Invariance**. To assess the importance of enforcing invariance over background removal and object styles, we extend SimCLR to use saliency masking and 4 large views and 2 small views, and compare it to RELICv2. We train

|  | Top-1 |
|---|---|
| SimCLR [Chen et al., 2020a] | 64.5% |
| SimCLR + [4, 2] + saliency | 66.2% |
| RELIC [Mitrovic et al., 2021] | 66.2% (+1.7%) |
| RELICv2 (ours) | 67.5% (+6.4%) |

both methods for 100 epochs and report top-1 accuracy on ImageNet. As we can see from the table, invariance-based methods profit from multiple views and saliency masking significantly more than purely contrastive methods. Thus, invariance plays a crucial role in learning better representations.

# 4 ADDITIONAL EXPERIMENTAL RESULTS

In addition to outperforming the supervised baseline across a wide range of different ResNet architectures on ImageNet classification, RELICv2 also achieves state-of-the-art performance on a wide range of downstream tasks and datasets. We showcase the generality and wide applicability of RELICv2 on transfer, semi-supervised, robustness and out-of-distribution (OOD) generalization tasks when pretraining the representations on ImageNet, but also on the much larger and more complex Joint Foto Tree (JFT-300M) dataset [Hinton et al., 2015; Chollet, 2017; Sun et al., 2017]. We report here the results on the transfer, robustness and OOD generalization downstream tasks and on the JFT-300M dataset. For a complete set of results, in particular in the semi-supervised setting and on JFT-300M, and experimental details refer to the appendix.

## 4.1 ROBUSTNESS AND OOD GENERALIZATION

We evaluate the robustness and out-of-distribution (OOD) generalization of RELICv2 on a wide variety of datasets. We use ImageNetV2 [Recht et al., 2019] and ImageNet-C [Hendrycks and Dietterich, 2019] to evaluate robustness, while ImageNet-R [Hendrycks et al., 2021], ImageNet-Sketch [Wang et al., 2019] and ObjectNet [Barbu et al., 2019] are used to evaluate OOD generalization. We evaluate the representations of a standard ResNet50 encoder under linear evaluation akin to 3, i.e. we train a linear classifier on top of the frozen representation using the labelled ImageNet training set; the test evaluation is performed zero-shot, i.e no training is done on the above datasets. In table 3, we see that RELICv2 learns more robust representations outperforming both supervised performance and competing self-supervised methods on ImageNetV2 and ImageNet-C. RELICv2 also learns representations that outperform competing self-supervised methods while being on par with supervised performance in terms of OOD generalization.

Table 3: Top-1 Accuracy (in %) under linear evaluation on ImageNetV2 and ImageNet-C (robustness), and ImageNet-R (IN-R), ImageNet-Sketch (IN-S) and ObjectNet (out-of-distribution). ImageNetv2 has three variants – matched frequency (MF), Threshold 0.7 (T-0.7) and Top Images (TI). The results for ImageNet-C (IN-C) are averaged across the 15 different corruptions.

|  | Robustness | | | | OOD Generalization | | |
|---|---|---|---|---|---|---|---|
| Method | MF | T-0.7 | Ti | IN-C | IN-R | IN-S | ObjectNet |
| Supervised | 65.1 | 73.9 | 78.4 | 40.9 | 24.0 | 6.1 | 26.6 |
| SimCLR [Chen et al., 2020a] | 53.2 | 61.7 | 68.0 | 31.1 | 18.3 | 3.9 | 14.6 |
| BYOL [Grill et al., 2020] | 62.2 | 71.6 | 77.0 | 42.8 | 23.0 | 8.0 | 23.0 |
| RELIC [Mitrovic et al., 2021] | 63.1 | 72.3 | 77.7 | 44.5 | 23.8 | 9.1 | 23.8 |
| RELICv2 (ours) | **65.3** | **74.5** | **79.4** | **44.8** | **23.9** | **9.9** | **25.9** |

## 4.2 TRANSFER TO OTHER CLASSIFICATION DATASETS AND SEMANTIC SEGMENTATION

**Classification.** We perform linear evaluation and fine-tuning on a wide range of classification benchmarks. We follow established evaluation protocols from the literature as described in the ap-

pendix and report standard metrics for each dataset on held-out test sets. Figure 5 compares the transfer performance of BYOL, NNCLR and RELICv2 pre-trained representations relative to supervised pre-training. RELICv2 improves upon both the supervised baseline and competing methods, performing best on 7 out of 11 tasks. RELICv2 has an average relative improvement of over 5% across all tasks—over double that of closest competing self-supervised method NNCLR. For detailed results on linear evaluation and fine-tuning see appendix.

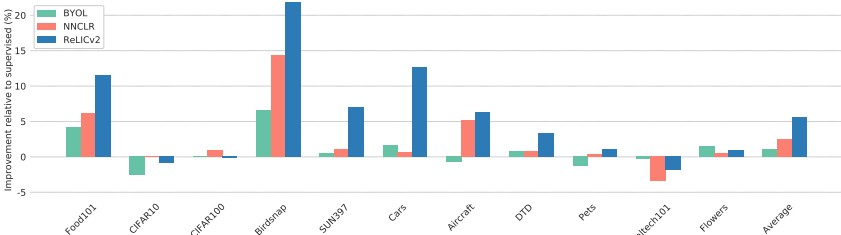

Figure 5: Transfer performance relative to the supervised baseline (a value of 0 indicates equal performance to supervised).

**Semantic Segmentation.** We evaluate the ability of RELICv2 to facilitate successful transfer of the learned representations to the PASCAL [Everingham et al., 2010] and Cityscapes [Cordts et al., 2016] semantic segmentation tasks. In accordance with [He et al., 2019], we use the RELICv2 ImageNet representation to initialise a fully convolutional backbone, which we fine-tune on the PASCAL `train_aug2012` set for 45 epochs and report the mean intersection over union (mIoU) on the `val2012` set. The fine-tuning on Cityscapes is done on the `train_fine` set for 160 epochs and evaluated on the `val_fine` set. From table 4 we see that RELICv2 significantly outperforms BYOL on both datasets, and also provides an improvement over the original ReLIC method. Interestingly, RELICv2 outperforms even DetCon [Hénaff et al., 2021], a method specifically trained for detection, on PASCAL where it achieves 77.3 (mIoU).

Table 4: Transfer performance on semantic segmentation tasks.

| Method | PASCAL | Cityscapes |
|---|---|---|
| BYOL [Grill et al., 2020] | 75.7 | 74.6 |
| ReLIC [Mitrovic et al., 2021] | 77.7 | 75.0 |
| ReLICv2 (ours) | **77.9** | **75.2** |

### 4.3 PRETRAINING ON JOINT FOTO TREE (JFT-300M)

We also test how well RELICv2 scales to much larger datasets by pretraining representations using the Joint Foto Tree (JFT-300M) dataset which consists of 300 million images from more than 18k classes [Hinton et al., 2015; Chollet, 2017; Sun et al., 2017]. We then evaluate the learned

Table 5: Top-1 accuracy (in %) on ImageNet when learning representations using the JFT-300M dataset.

| Method | Top-1 | | |
|---|---|---|---|
| | 1000 epochs | 3000 epochs | >4000 epochs |
| ReLIC [Mitrovic et al., 2021] | 61.5 | 61.5 | – |
| BYOL [Grill et al., 2020] | 67.0 | 67.6 | 67.9 (5000 epochs) |
| Divide and Contrast [Tian et al., 2021] | 67.9 | 69.8 | 70.7 (4500 epochs) |
| ReLICv2 (ours) | **70.3** | **71.1** | **71.4** (5000 epochs) |

representations on the ImageNet test set under the same linear evaluation protocol as described in section 3. We compare RELICv2 against BYOL and Divide and Contrast [Tian et al., 2021], a method that was specifically designed to handle large and uncurated datasets and represents the current state-of-art in self-supervised JFT-300M pretraining. Table 5 reports the top-1 accuracy when training the various methods using the standard ResNet50 architecture as the backbone for different number of ImageNet equivalent epochs on JFT-300M. RELICv2 improves over Divide and Contrast [Tian et al., 2021] by more than 2% when training on JFT for 1000 epochs and achieves better overall performance than competing methods while needing a smaller number of training epochs. For experimental details and additional results on JFT-300M, refer to the appendix.

## 5 RELATED WORK

Recently, *contrastive* multi-view approaches have become an important area of research owing to their excellent performance in visual recognition tasks [Oord et al., 2018; Bachman et al., 2019; Chen et al., 2020a; He et al., 2019; Dwibedi et al., 2021; Grill et al., 2020]. Moreover, bootstrapping-based multi-view learning has also achieved comparable performance [Grill et al., 2020]. The aforementioned methods implicitly enforce invariance by maximizing the similarity between views. On

the other hand, [Caron et al., 2020] incorporates an explicit clustering step as a more direct way of enforcing some notion of invariance. However, neither of these strategies can be directly linked theoretically to learning more compact representations. [Mitrovic et al., 2021] approach invariance from a causal perspective and use an explicit invariance loss in conjunction with a contrastive objective to learn representations. They show that invariance must be explicitly enforced—via an invariance loss in addition to a contrastive loss—in order to obtain guaranteed generalization performance.

Background augmentations have recently been proposed as part of the data augmentation pipeline in [Zhao et al., 2021] and further discussed in concurrent work to our manuscript [Ryali et al., 2021a;b]. In particular, [Ryali et al., 2021a;b] obtain saliency masks by training an unsupervised saliency detection network using the self-supervised refinement mechanism from DeepUSPS [Nguyen et al., 2019]. While [Ryali et al., 2021a;b] use an additional dataset, MSRA-B [Liu et al., 2010], to train their saliency network, we use a subset of ImageNet to learn the saliency network thus enabling a fair comparison to prior work. RELICv2 differs from these methods in three key points. First, we enforce an explicit invariance loss between a view with and without the background, while [Zhao et al., 2021; Ryali et al., 2021a;b] do not have any invariance loss. Second, RELICv2 learns representations using multiple views of varying sizes, while these methods use only two large views. Third, unlike in [Zhao et al., 2021; Ryali et al., 2021a;b] the focus of our paper is to explore whether it is possible to achieve better than supervised performance on a wide range of ResNet architectures, rather than to just investigate the benefits of background augmentations combined with a number of self-supervised methods on just the ResNet 50 1x encoder. To learn the spatial structure of the image and as a computationally favourable alternative to large views, [Caron et al., 2020] propose to use multiple smaller views in addition to the two large views. In a notable difference to [Caron et al., 2020], RELICv2 employs small views to increase the representation robustness and proposes to use only a small number of small views. Additionally, RELICv2 imposes an invariance loss between pairs of large and small views, and uses more large views. Outside of contrastive learning, [Lee et al., 2021] take a conditional entropy bottleneck approach which results in better-than-supervised performance albeit only on the ResNet50 (2x) architecture (see figure 1). See appendix for more detailed comparison of related works.

## 6 DISCUSSION

We proposed a novel self-supervised representation learning method RELICv2 which learns representations by enforcing invariance over multiple different views of varying sizes and saliency masking. This enables us to learn robust representations which are more invariant to the above spurious features as evidenced by substantial improvement over existing state-of-the-art in our extensive experimental analysis across a wide range of downstream settings, tasks and datasets, In particular, RELICv2 is the new state-of-the-art in linear evaluation, transfer learning and robustness on ImageNet across ResNet-based self-supervised methods. Moreover, RELICv2 is the first method that demonstrates that representations learned without access to labels can consistently outperform a standard supervised baseline on ImageNet which is a first step in surpassing supervised learning.

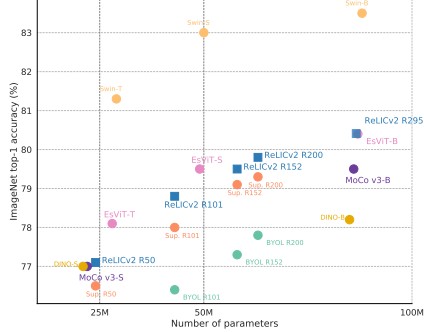

Figure 6: Comparison of ImageNet top-1 accuracy between RELICv2 and recent vision transformer-based architectures (Swin [Liu et al., 2021] is a fully supervised transformer baseline).

Vision transformers (ViTs) [Dosovitskiy et al., 2020] have recently emerged as promising architectures for visual representation learning. Figure 6 compares recent ViT-based methods against RELICv2 using a variety of larger ResNet architectures. Notably, RELICv2 outperforms recent self-supervised ViT-based methods DINO [Caron et al., 2021] and MoCov3 [Chen et al., 2021] as well as exhibiting similar performance to EsViT [Li et al., 2021] for comparable parameter counts despite these methods using more powerful architectures and more involved training procedures. Our results suggest that combining the insights we have developed with RELICv2 alongside recent architectural innovations could lead to further improvements in representation learning and more powerful foundation models.

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

# A   COMPARISON BETWEEN SELF-SUPERVISED METHODS

In this review we focus on how important algorithmic choices: namely explicitly enforcing invariance and more considered treatment of positive and negative examples are key factors in improving downstream classification performance of unsupervised representations.

| Method | Contrastive | Invariance | Positives | Negatives |
|---|---|---|---|---|
| SimCLR [Chen et al., 2020a] | ✓ | ✗ | $\text{aug}(x_i)$ | full batch |
| BYOL [Grill et al., 2020] | ✗ | $\ell_2$ | $\text{aug}(x_i)$ | n/a |
| NNCLR [Dwibedi et al., 2021] | ✓ | ✗ | $\text{aug}(x_i), \text{nn}(x_i)$ | full batch |
| MoCo [He et al., 2019] | ✓ | ✗ | $\text{aug}(x_i)$ | queue |
| SwAV [Caron et al., 2020] | ✓ | ✗ | $\text{aug}(x_i), \text{mc}(x_i), \text{proto}^+(x_i)$ | $\text{proto}^-(x_i)$ |
| Debiased [Chuang et al., 2020] | ✓ | ✗ | $\text{aug}(x_i)$ | importance sample |
| Hard Negatives [Robinson et al., 2020] | ✓ | ✗ | $\text{aug}(x_i)$ | importance sample |
| ReLICv1 [Mitrovic et al., 2021] | ✓ | $D_{\text{KL}}$ | $\text{aug}(x_i)$ | subsample |
| RELICv2 (ours) | ✓ | $D_{\text{KL}}$ | $\text{aug}(x_i), \text{mc}(x_i), \text{sal}(x_i)$ | subsample |

Table 6: The role of positives and negatives in recent unsupervised representation learning algorithms.

**Negatives.**   A key observation of [Chen et al., 2020a] was that large batches (up to 4096) improve results. This was partly attributed to the effect of more negatives. This motivated the incorporation of queues that function as large reservoirs of negative examples into contrastive learning [He et al., 2019]. However subsequent work has shown that naively using a large number of negatives can have a detrimental effect on learning [Mitrovic et al., 2020; Saunshi et al., 2019; Chuang et al., 2020; Robinson et al., 2020]. One reason for this is due to *false negatives*, that is points in the set of negatives which actually belong to the same latent class as the anchor point. These points are likely to have a high relative similarity to the anchor under $\phi$ and therefore contribute disproportionately to the loss. This will have the effect of pushing apart points belonging to the same class in representation space. The selection of true negatives is a difficult problem as in the absence of labels it necessitates having access to reasonably good representations to begin with. As we do not have access to these representations, but are instead trying to learn them, there has been limited success in avoiding false and selecting informative negatives. This phenomenon explains the limited success of attempts to perform hard negative sampling.

Subsampling-based approaches have been proposed to avoid false negatives via importance sampling to attempt to find *true* negatives which are close to the latent class boundary of the anchor point [Robinson et al., 2020], or uniformly-at-random sampling a small number of points to avoid false negatives [Mitrovic et al., 2020].

**Positives and invariance.**   Learning representations which are invariant to data augmentation is known to be important for self-supervised learning. Invariance is achieved heuristically through comparing two different augmentations of the same anchor point. Incorporating an explicit clustering step is another way of enforcing some notion of invariance [Caron et al., 2020]. However, neither of these strategies can be directly linked theoretically to learning more compact representations. More rigorously [Mitrovic et al., 2021] approach invariance from a causal perspective. They show that invariance must be explicitly enforced—via an invariance loss in addition to the contrastive loss—in order to obtain guaranteed generalization performance. Most recently [Dwibedi et al., 2021] and [Assran et al., 2021] use nearest neighbours to identify other elements from the batch which potentially belong to the same class as the anchor point.

Table 6 provides a detailed comparison in terms of how prominent representation learning methods utilize positive and negative examples and how they incorporate both explicit contrastive and invari-

ance losses. Here $\text{aug}(x_i)$ refers to the standard set of SimCLR augmentations [Chen et al., 2020a], $\text{nn}(x_i)$ refers to a scheme which selects nearest neighbours of $x_i$, $\text{mc}(x_i)$ are multicrop augmentations (c.f. [Caron et al., 2020]). $\text{proto}^+(x_i)$ and $\text{proto}^-(x_i)$ refer to using prototypes computed via an explicit clustering step c.f. [Caron et al., 2020]. Finally, $\text{sal}(x_i)$ refers to a scheme which computes saliency masks of $x_i$ and removes backgrounds as described in section 2. Note that SwAV first computes a clustering of the batch then contrasts the embedding of the point and its nearest cluster centroid ($\text{proto}^+$) against the remaining $K-1$ cluster centroids ($\text{proto}^-$); invariance is implicitly enforced in the clustering step.

# B  PRETRAINING ON IMAGENET – IMPLEMENTATION DETAILS AND ADDITIONAL RESULTS

Similar to previous work [Chen et al., 2020a; Grill et al., 2020] we minimize our objective using the LARS optimizer [You et al., 2017] with a cosine decay learning rate schedule without restarts. Unless otherwise indicated, we train our models for 1000 epochs with a warm-up period of 10 epochs and a batch size of $|\mathcal{B}| = 4096$. In our experiments, we use 4 views of the standard size $224 \times 224$ and 2 views of the smaller size $96 \times 96$ each coming from an image augmented by a different randomly chosen data augmentation; the smaller size views are centered crops of the randomly augmented image. For a detailed ablation analysis on the number of large and small crops see section 3.1.

## B.1  LINEAR EVALUATION

Following the approach of [Chen et al., 2020a; Grill et al., 2020; Caron et al., 2020; Dwibedi et al., 2021], we use the standard linear evaluation protocol on ImageNet. We train a linear classifier on top of the frozen representation which has been pretrained, i.e. the encoder parameters as well as the batch statistics are not being updated. For training the linear layer, we preprocess the data by applying standard spatial augmentations, i.e. randomly cropping the image with subsequent resizing to $224 \times 224$ and then randomly applying a horizontal flip. At test time, we resize images to 256 pixels along the shorter side with bicubic resampling and apply a $224 \times 224$ center crop to it. Both for training and testing, after performing the above processing, we normalize the color channels by substracting the average channel value and dividing by the standard deviation of the channel value (as computed on ImageNet). To train the linear classifier, we optimize the cross-entropy loss with stochastic gradient descent with Nestorov momentum for 100 epochs using a batch size of 1024 and a momentum of 0.9; we do not use any weight decay or other regularization techniques.

In the following tables, we report the top-1 and top-5 accuracies of different methods under a varied set of ResNet encoders of different sizes, spanning ResNet50, ResNet101, ResNet152 and ResNet200 and layer widths of $1\times$, $2\times$ and $4\times$. ResNet50 with $2\times$ and $4\times$ wider layers has 94 and 375 million parameters, respectively. ResNet101, ResNet152, ResNet200 and ResNet200 $2\times$ have 43, 58, 63 and 250 million parameters, respectively.

In table 8, we present results under linear evaluation on the ImageNet test set a varied set of ResNet architectures; we compare against different unsupervised representation learning methods and use as the supervised baselines the results reported in [Chen et al., 2020a; Grill et al., 2020]. Note that the supervised baselines reported in [Chen et al., 2020a] are extensively used throughout the self-supervised

| Method | Top-1 | Top-5 |
|---|---|---|
| Supervised [Chen et al., 2020a] | 76.5 | 93.7 |
| SimCLR [Chen et al., 2020a] | 69.3 | 89.0 |
| MoCo v2 [Chen et al., 2020b] | 71.1 | - |
| InfoMin Aug. [Tian et al., 2020] | 73.0 | 91.1 |
| BYOL [Grill et al., 2020] | 74.3 | 91.6 |
| RELIC [Mitrovic et al., 2021] | 74.8 | 92.2 |
| SwAV [Caron et al., 2020] | 75.3 | - |
| NNCLR [Dwibedi et al., 2021] | 75.6 | 92.4 |
| C-BYOL [Lee et al., 2021] | 75.6 | 92.7 |
| RELICv2 (ours) | **77.1** | 93.3 |

Table 7: Top-1 and top-5 accuracy (in %) under linear evaluation on the ImageNet test set for a ResNet50 encoder set for different representation learning methods.

literature in order to compare performance against supervised learning. For architectures for which supervised baselines are not available in [Chen et al., 2020a], we use supervised baselines reported in [Grill et al., 2020] which use stronger augmentations for training supervised models than [Chen et al., 2020a] and as such do not represent a direct like-for-like comparison with self-supervised methods. Across this varied set of ResNet architectures, RELICv2 outperforms all competing self-supervised methods while also outperforming the supervised baselines in all cases with margins up to 1.2% in absolute terms.

| Method | Top-1 | Top-5 |
|---|---|---|
| Supervised [Chen et al., 2020a] | 77.8 | – |
| MoCo [He et al., 2019] | 65.4 | – |
| SimCLR [Chen et al., 2020a] | 74.2 | 92.0 |
| BYOL [Grill et al., 2020] | 77.4 | 93.6 |
| SwAV [Caron et al., 2020] | 77.3 | – |
| C-BYOL [Lee et al., 2021] | 78.8 | **94.5** |
| RELICv2 (ours) | **79.0** | **94.5** |

(a) ResNet50 $2\times$ encoder.

| Method | Top-1 | Top-5 |
|---|---|---|
| Supervised [Chen et al., 2020a] | 78.9 | – |
| MoCo [He et al., 2019] | 68.6 | – |
| SimCLR [Chen et al., 2020a] | 76.5 | 93.2 |
| SwAV [Caron et al., 2020] | 77.9 | – |
| BYOL [Grill et al., 2020] | 78.6 | 94.2 |
| RELICv2 (ours) | **79.4** | **94.3** |

(b) ResNet50 $4\times$ encoder.

| Method | Top-1 | Top-5 |
|---|---|---|
| Supervised [Grill et al., 2020] | 78.0 | 94.0 |
| BYOL [Grill et al., 2020] | 76.4 | 93.0 |
| RELICv2 (ours) | **78.7** | **94.4** |

(c) ResNet101 encoder.

| Method | Top-1 | Top-5 |
|---|---|---|
| Supervised [Grill et al., 2020] | 79.1 | 94.5 |
| BYOL [Grill et al., 2020] | 77.3 | 93.7 |
| RELICv2 (ours) | **79.3** | **94.6** |

(d) ResNet152 encoder.

| Method | Top-1 | Top-5 |
|---|---|---|
| Supervised [Grill et al., 2020] | 79.3 | 94.6 |
| BYOL [Grill et al., 2020] | 77.8 | 93.9 |
| RELICv2 (ours) | **79.8** | **95.0** |

(e) ResNet200 encoder.

| Method | Top-1 | Top-5 |
|---|---|---|
| Supervised [Grill et al., 2020] | 80.1 | **95.2** |
| BYOL [Grill et al., 2020] | 79.6 | 94.8 |
| RELICv2 (ours) | **80.6** | **95.2** |

(f) ResNet200 $2\times$ encoder.

Table 8: Top-1 and top-5 accuracy (in %) under linear evaluation on the ImageNet test set for a varied set of ResNet architectures.

## B.2 SEMI-SUPERVISED LEARNING

We further test RELICv2 representations learned on bigger ResNet models in the semi-supervised setting. For this, we follow the semi-supervised protocol as in [Zhai et al., 2019; Chen et al., 2020a; Grill et al., 2020; Caron et al., 2020]. First, we initialize the encoder with the parameters of the pretrained representation and we add on top of this encoder a linear classifier which is randomly initialized. Then we train both the encoder and the linear layer using either 1% or 10% of the ImageNet training data; for this we use the splits introduced in [Chen et al., 2020a] which have been used in all the methods we compare to [Grill et al., 2020; Caron et al., 2020; Dwibedi et al., 2021; Lee et al., 2021]. For training, we randomly crop the image and resize it to $224 \times 224$ and then randomly apply a horizontal flip. At test time, we resize images to 256 pixels along the shorter side with bicubic resampling and apply a $224 \times 224$ center crop to it. Both for training and testing, after performing the above processing, we normalize the color channels by substracting the average channel value and dividing by the standard deviation of the channel value (as computed on ImageNet). Note that this is the same data preprocessing protocol as in the linear evaluation protocol. To train the model, we use a cross entropy loss with stochastic gradient descent with Nesterov momentum of 0.9. For both 1% and 10% settings, we train for 20 epochs and decay the initial learning rate by a factor 0.2 at 12 and 16 epochs. Following the approach of [Caron et al., 2020], we use the optimizer with different learning rates for the encoder and linear classifier parameters. For the 1% setting, we use a batch size of 2048 and base learning rates of 10 and 0.04 for the linear layer and encoder, respectively; we do not use any weight decay or other regularization technique. For the 10% setting, we use a batch size of 512 and base learning rates of 0.3 and 0.004 for the linear layer and encoder, respectively; we use a weight decay of $1e - 5$, but do not use any other regularization technique.

Top-1 and top-5 accuracy on the ImageNet test set is reported in table 9. RELICv2 outperforms both the standard supervised baseline and all previous state-of-the-art self-supervised methods when using 10% of the data for fine-tuning. When using 1% of the data, only C-BYOL performs better than RELICv2. From table 10, we see that RELICv2 outperforms competing self-supervised methods on ResNet50 $2\times$ in both the 1% and 10% setting. For larger ResNets, ResNet50 $4\times$ and ResNet200 $2\times$, RELICv2 is state-of-the-art with respect to top-1 accuracy for the low-data regime of 1%. On

| Method | Top-1 | | Top-5 | |
|---|---|---|---|---|
| | 1% | 10% | 1% | 10% |
| Supervised [Zhai et al., 2019] | 25.4 | 56.4 | 48.4 | 80.4 |
| SimCLR [Chen et al., 2020a] | 48.3 | 65.6 | 75.5 | 87.8 |
| BYOL [Grill et al., 2020] | 53.2 | 68.8 | 78.4 | 89.00 |
| SwAV [Caron et al., 2020] | 53.9 | 70.2 | 78.5 | 89.9 |
| NNCLR [Dwibedi et al., 2021] | 56.4 | 69.8 | 80.7 | 89.3 |
| C-BYOL [Lee et al., 2021] | **60.6** | 70.5 | **83.4** | 90.0 |
| RELICv2 (ours) | 58.1 | **72.4** | 81.3 | **91.2** |

Table 9: Top-1 and top-5 accuracy (in %) on the ImageNet test set after semi-supervised training with a fraction of ImageNet labels on a ResNet50 encoder for different representation learning methods.

these networks for the higher data regime of $10\%$ BYOL outperforms RELICv2. Note that BYOL trains their semi-supervised models for 30 or 50 epochs whereas RELICv2 is trained only for 20 epochs. We hypothesize that longer training (e.g. 30 or 50 epochs as BYOL) is needed for RELICv2 representations on larger ResNets as there are more model parameters.

| Method | Top-1 | | Top-5 | |
|---|---|---|---|---|
| | 1% | 10% | 1% | 10% |
| *ResNet50 2× encoder* | | | | |
| SimCLR [Chen et al., 2020a] | 58.5 | 71.7 | 83.0 | 91.2 |
| BYOL [Grill et al., 2020] | 62.2 | 73.5 | 84.1 | 91.7 |
| RELICv2 (ours) | **64.7** | **73.7** | **85.4** | **92.0** |
| *ResNet50 4× encoder* | | | | |
| SimCLR [Chen et al., 2020a] | 63.0 | 74.4 | 85.8 | 92.6 |
| BYOL [Grill et al., 2020] | 69.1 | **75.7** | **87.9** | **92.5** |
| RELICv2 (ours) | **69.5** | 74.6 | 87.3 | 91.6 |
| *ResNet200 2× encoder* | | | | |
| BYOL [Grill et al., 2020] | 71.2 | **77.7** | **89.5** | **93.7** |
| RELICv2 (ours) | **72.1** | 76.4 | **89.5** | 93.0 |

Table 10: Top-1 and top-5 accuracy (in %) after semi-supervised training with a fraction of ImageNet labels for different ResNet encoders and unsupervised representation learning methods. Results are reported on the ImageNet test set.

## B.3 TRANSFER

We follow the transfer performance evaluation protocol as outlined in [Grill et al., 2020; Chen et al., 2020a]. We evaluate RELICv2 in both transfer settings – linear evaluation and fine-tuning. For the linear evaluation protocol we freeze the encoder and train only a randomly initialized linear classifier which is put on top of the encoder. On the other hand, for fine-tuning in addition to training the randomly initialized linear classifier, we also allow for gradients to propagate to the encoder which has been initialized with the parameters of the pretrained representation. In line with prior work [Chen et al., 2020a; Grill et al., 2020; Dwibedi et al., 2021], we test RELICv2 representations on the following datasets: Food101 [Bossard et al., 2014], CIFAR10 [Krizhevsky et al., 2009], CIFAR100 [Krizhevsky et al., 2009], Birdsnap [Berg et al., 2014], SUN397 (split 1) [Xiao et al., 2010], DTD (split 1) [Cimpoi et al., 2014], Cars [Krause et al., 2013] Aircraft [Maji et al., 2013], Pets [Parkhi et al., 2012], Caltech101 [Fei-Fei et al., 2004], and Flowers [Nilsback and Zisserman, 2008].

Again in line with previous methods [Chen et al., 2020a; Grill et al., 2020; Dwibedi et al., 2021], for Food101 [Bossard et al., 2014], CIFAR10 [Krizhevsky et al., 2009], CIFAR100 [Krizhevsky et al., 2009], Birdsnap [Berg et al., 2014], SUN397 (split 1) [Xiao et al., 2010], DTD (split 1) [Cimpoi et al., 2014], and Cars [Krause et al., 2013] we report the Top-1 accuracy on the test set, and for

Aircraft [Maji et al., 2013], Pets [Parkhi et al., 2012], Caltech101 [Fei-Fei et al., 2004], and Flowers [Nilsback and Zisserman, 2008] we report the mean per-class accuracy as the relevant metric in the comparisons. For DTD and SUN397, we only use the first split, of the 10 provided splits in the dataset as per [Chen et al., 2020a; Grill et al., 2020; Dwibedi et al., 2021].

We train on the training sets of the individual datasets and sweep over different values of the models hyperparameters. To select the best hyperparameters, we use the validation sets of the individual datasets. Using the chosen hyperparameters, we train the appropriate using the merged training and validation data and test on the held out test data in order to obtain the numbers reported in table 11. We swept over learning rates $\{.01, 0.1, 0.2, 0.25, 0.3, 0.35, 0.4, 1., 2.\}$, batch sizes $\{128, 256, 512, 1024\}$, weight decay between $\{1e{-}6, 1e{-}5, 1e{-}4, 1e{-}3, 0.01, 0.1\}$, warmup epochs $\{0, 10\}$, momentum $\{0.9, 0.99\}$, Nesterov $\{$True, False$\}$, and the number of training epochs. For linear transfer we considered setting epochs among $\{20, 30, 60, 80, 100\}$, and for fine-tuning, we also considered $\{150, 200, 250\}$, for datasets where lower learning rates were preferable. Models were trained with the SGD optimizer with momentum.

As can be seen from table 11, RELICv2 representations yield better performance than both state-of-the-art self-supervised method as well as the supervised baseline across a wide range of datasets. Specifically, RELICv2 is best on 7 out of 11 datasets and on 8 out of 11 datasets in the linear and fine-tuning settings, respectively.

| Method | Food101 | CIFAR10 | CIFAR100 | Birdsnap | SUN397 | Cars | Aircraft | DTD | Pets | Caltech101 | Flowers |
|---|---|---|---|---|---|---|---|---|---|---|---|
| *Linear evaluation:* | | | | | | | | | | | |
| Supervised-IN [Chen et al., 2020a] | 72.3 | 93.6 | 78.3 | 53.7 | 61.9 | 66.7 | 61.0 | 74.9 | 91.5 | **94.5** | 94.7 |
| SimCLR [Chen et al., 2020a] | 68.4 | 90.6 | 71.6 | 37.4 | 58.8 | 50.3 | 50.3 | 74.5 | 83.6 | 90.3 | 91.2 |
| BYOL [Grill et al., 2020] | 75.3 | 91.3 | 78.4 | 57.2 | 62.2 | 67.8 | 60.6 | 75.5 | 90.4 | 94.2 | **96.1** |
| NNCLR [Dwibedi et al., 2021] | 76.7 | **93.7** | **79.0** | 61.4 | 62.5 | 67.1 | 64.1 | 75.5 | 91.8 | 91.3 | 95.1 |
| ReLICv2 (ours) | **80.6** | 92.8 | 78.2 | **65.4** | 66.2 | **75.1** | 64.8 | **77.4** | 92.4 | 92.8 | 95.6 |
| *Fine-tuned:* | | | | | | | | | | | |
| Random Init [Chen et al., 2020a] | 86.9 | 95.9 | 80.2 | 76.1 | 53.6 | 91.4 | 85.9 | 64.8 | 81.5 | 72.6 | 92.0 |
| Supervised-IN [Chen et al., 2020a] | 88.3 | 97.5 | **86.4** | 75.8 | 64.3 | 92.1 | 86.0 | 74.6 | 92.1 | 93.3 | 97.6 |
| SimCLR [Chen et al., 2020a] | 88.2 | 97.7 | 85.9 | 75.9 | 63.5 | 91.3 | 88.1 | 73.2 | 89.2 | 92.1 | 97.0 |
| BYOL [Grill et al., 2020] | 88.5 | **97.8** | 86.1 | 76.3 | 63.7 | 91.6 | 88.1 | 76.2 | 91.7 | **93.8** | 97.0 |
| ReLICv2 (ours) | **88.7** | 97.7 | 85.3 | **76.7** | 64.7 | 92.3 | 88.7 | 76.9 | 92.2 | 93.2 | **97.9** |

Table 11: Accuracy (in %) of transfer performance of a ResNet50 pretrained on ImageNet.

### B.4 ROBUSTNESS AND OOD GENERALIZATION

The robustness and out-of-distribution (OOD) generalization abilities of RELICv2 representations are tested on several detasets. We use ImageNetV2 [Recht et al., 2019] and ImageNet-C [Hendrycks and Dietterich, 2019] datasets to evaluate robustness. ImageNetV2 [Recht et al., 2019] has three sets of 10000 images that were collected to have a similar distribution to the original ImageNet validation set, while ImageNet-C [Hendrycks and Dietterich, 2019] consists of 15 synthetically generated corruptions (e.g. blur, noise) that are added to the ImageNet validation set.

For OOD generalization we examine the performance on ImageNet-R [Hendrycks et al., 2021], ImageNetSketch [Wang et al., 2019] and ObjectNet [Barbu et al., 2019]. ImageNet-R [Hendrycks et al., 2021] consists of 30000 different renditions (e.g. paintings, cartoons) of 200 ImageNet classes, while ImageNet-Sketch [Wang et al., 2019] consists of 50000 images, 50 for each ImageNet class, of object sketches in the black-and-white color scheme. These datasets aim to test robustness to different textures and other naturally occurring style changes and are out-of-distribution to the ImageNet training data. ObjectNet [Barbu et al., 2019] has 18574 images from differing viewpoints and backgrounds compared to ImageNet.

On all datasets we evaluate the representations of a standard ResNet50 encoder under a linear evaluation protocol akin to Section 3, i.e. we freeze the pretrained representations and train a linear classifier using the labelled ImageNet training set; the test evaluation is performed zero-shot, i.e no training is done on the above datasets. As we have seen in table 3, RELICv2 learns more robust representations and outperforms both the supervised baseline and the competing self-supervised

methods on ImageNetV2 and ImageNet-C. We provide a detailed breakdown across the different ImageNet-C corruptions in table 12.

| Method | Gauss | Shot | Impulse | Defocus | Blur Glass | Motion | Zoom | Snow | Weather Frost | Fog | Bright | Contrast | Digital Elastic | Pixel | JPEG |
|---|---|---|---|---|---|---|---|---|---|---|---|---|---|---|---|
| Supervised [Lim et al., 2019] | 37.1 | 35.1 | 30.8 | 36.8 | **25.9** | 34.9 | **38.1** | 34.5 | 40.7 | 56.9 | 68.1 | 40.6 | **45.6** | 32.6 | 56.0 |
| SimCLR [Chen et al., 2020a] | 29.1 | 26.3 | 17.3 | 22.1 | 14.7 | 20.0 | 18.6 | 27.2 | 33.3 | 46.2 | 59.7 | 53.9 | 31.0 | 24.2 | 43.9 |
| BYOL [Grill et al., 2020] | 41.5 | 38.7 | 31.9 | 37.8 | 22.5 | 31.6 | 29.6 | 35.1 | 42.9 | 60.1 | 69.0 | 58.4 | 41.5 | 46.3 | 55.9 |
| RELIC [Mitrovic et al., 2021] | **43.4** | **40.7** | **36.6** | **40.5** | 24.5 | 34.3 | 30.5 | 36.6 | 43.8 | 61.4 | 69.5 | 59.5 | 42.8 | **46.8** | 57.3 |
| RELICv2 (ours) | 41.6 | 39.0 | 31.1 | 39.7 | 22.6 | **35.2** | 34.5 | **40.1** | **46.1** | **64.5** | **71.0** | **60.0** | 44.6 | 46.6 | **58.4** |

Table 12: Top-1 accuracies for for Gauss, Shot, Impulse, Blur, Weather, and Digital corruption types on ImageNet-C.

# C ANALYSIS

**Class confusion.** To understand the effect of the invariance term in RELICv2, we look at the distances between learned representations of closely related classes. Figure 7 illustrates the Euclidean distances between nearest-neighbour representations learned by RELICv2 and BYOL on ImageNet using the protocol described in section 4. Here we pick two breeds of dog and two breeds of cat. Each of these four classes has 50 points associated with it from the ImageNet validation set, ordered contiguously. Each row represents an image and each coloured point in a row represents one of the five nearest neighbours of the representation of that image where the colour indicates the distance between the image and the nearest neighbour. Representations which align perfectly with the underlying class structure would exhibit a perfect block-diagonal structure, i.e. their nearest neighbours all belong to the same underlying class. We see that RELICv2 learns representations whose nearest neighbours are closer and exhibit less confusion between classes and super-classes than BYOL.

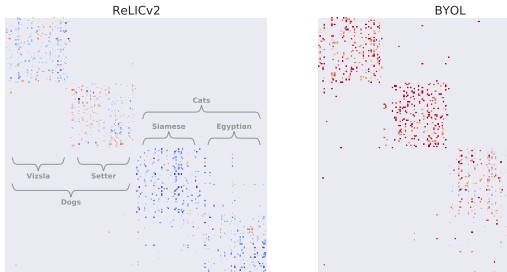

Figure 7: Distances between nearest-neighbour representations. Each coloured point in a row represents one of the five nearest neighbours of the representation of that image where the colour indicates the distance between the points.

# D PRETRAINING ON JOINT FOTO TREE (JFT-300M) – IMPLEMENTATION DETAILS AND ADDITIONAL RESULTS

## D.1 LINEAR EVALUATION

We test how well RELICv2 scales to much larger datasets by pretraining representations using the Joint Foto Tree (JFT-300M) dataset which consists of 300 million images from more than 18k classes [Hinton et al., 2015; Chollet, 2017; Sun et al., 2017]. We then evaluate the learned representations on the ImageNet validation set under the same linear evaluation protocol as described in section 3. We compare RELICv2 against BYOL and Divide and Contrast (DnC) [Tian et al., 2021], a method that was specifically designed to handle large and uncurated datasets and represents the current state-of-art in self-supervised JFT-300M pretraining. Table 13 reports the top-1 accuracy when training the various methods using the standard ResNet50 architecture as the backbone for different number of ImageNet equivalent epochs on JFT-300M; implementation details can be found in the supplementary material. RELICv2 improves over DnC by more than $2\%$ when training on JFT for 1000 epochs and achieves better overall performance than competing methods while needing a smaller number of training epochs.

| Method | Epochs | Top-1 |
|---|---|---|
| BYOL [Grill et al., 2020] | 1000 | 67.0 |
| Divide and Contrast [Tian et al., 2021] | 1000 | 67.9 |
| RELICv2 (ours) | 1000 | **70.3** |
| BYOL [Grill et al., 2020] | 3000 | 67.6 |
| Divide and Contrast [Tian et al., 2021] | 3000 | 69.8 |
| RELICv2 (ours) | 3000 | **71.1** |
| BYOL [Grill et al., 2020] | 5000 | 67.9 |
| Divide and Contrast [Tian et al., 2021] | 4500 | 70.7 |
| RELICv2 (ours) | 5000 | **71.4** |

Table 13: Top-1 accuracy (in %) on ImageNet when learning representations using the JFT-300M dataset. Each method is pre-trained on JFT-300M for an ImageNet-equivalent number of epochs and evaluted on the ImageNet validation set under a linear evaluation protocol.

For results reported in table 13, we use the following training and evaluation protocol. To pre-train RELICv2 on the Joint Foto Tree (JFT-300M) dataset, we used a base learning rate of $0.3$ for pretraining the representations for 1000 ImageNet-equivalent epochs. For longer pretraining of 3000 and 5000 ImageNet-equivalent epochs, we use a lower base learning rate of $0.2$. We set the target exponential moving average to $0.996$, the contrast scale to $0.3$, temperature to $0.2$ and the saliency mask apply probability to $0.15$ for all lenghts of pretraining. For 1000 and 5000 ImageNet-equivalent epochs we use $2.0$ as the invariance scale, while for 3000 ImageNet-equivalent epochs, we use invariance scale $1.0$. We then follow the linear evaluation protocol on ImageNet described in Appendix B.1. We train a linear classifier on top of the pretrained representations from JFT-300M with stochastic gradient descent with Nesterov momentum for 100 epochs using batch size of 256, learning rate of $0.5$ and momentum of $0.9$.

## D.2 TRANSFER

We evaluate the transfer performance of JFT-300M pretrained representations under the linear evaluation protocol. For this, we freeze the encoder and train only linear classifier on top of the frozen encoder output, i.e. representation. As before in B.3, we follow the transfer performance evaluation protocol as outlined in [Grill et al., 2020; Chen et al., 2020a]. In line with prior work, for Food101 [Bossard et al., 2014], CIFAR10 [Krizhevsky et al., 2009], CIFAR100 [Krizhevsky et al., 2009], Birdsnap [Berg et al., 2014], SUN397 (split 1) [Xiao et al., 2010], DTD (split 1) [Cimpoi et al., 2014], and Cars [Krause et al., 2013] we report the top-1 accuracy on the test set, and for Aircraft [Maji et al., 2013], Pets [Parkhi et al., 2012], Caltech101 [Fei-Fei et al., 2004], and Flowers [Nilsback and Zisserman, 2008] we report the mean per-class accuracy as the relevant metric in

the comparisons. For DTD and SUN397, we only use the first split, of the 10 provided splits in the dataset.

We train on the training sets of the individual datasets and sweep over different values of the models hyperparameters. To select the best hyperparameters, we use the validation sets of the individual datasets. Using the chosen hyperparameters, we train the linear layer from scratch using the merged training and validation data and test on the held out test data in order to obtain the numbers reported in table 14. We swept over learning rates $\{.01, 0.1, 0.2, 0.25, 0.3, 0.35, 0.4, 1., 2.\}$, batch sizes $\{128, 256, 512, 1024\}$, weight decay between $\{1e-6, 1e-5, 1e-4, 1e-3, 0.01, 0.1\}$, warmup epochs $\{0, 10\}$, momentum $\{0.9, 0.99\}$, Nesterov $\{True, False\}$, and the number of training epochs $\{60, 80, 100\}$. Models were trained with the SGD optimizer with momentum.

As can be seen from table 14, longer pretraining benefits transfer performance of RELICv2. Although DnC [Tian et al., 2021] was specifically developed to handle uncurated datasets such as JFT-300M, we see that RELICv2 has comparable performance to DnC in terms of the number of datasets with state-of-the-art performance among self-supervised representation learning methods; this showcases the generality of RELICv2.

| Method | Food101 | CIFAR10 | CIFAR100 | Birdsnap | SUN397 | Cars | Aircraft | DTD | Pets | Caltech101 | Flowers |
|---|---|---|---|---|---|---|---|---|---|---|---|
| BYOL-5k [Grill et al., 2020] | 73.3 | 89.8 | 72.4 | 38.2 | 61.8 | 64.4 | 54.4 | 75.5 | 77 | 90.1 | 94.3 |
| DnC-4.5k [Tian et al., 2021] | **78.7** | **91.7** | **74.9** | 42.1 | 65.0 | 75.3 | 54.1 | 76.6 | **86.1** | 90.2 | **98.2** |
| ReLICv2-1k (ours) | 77.5 | 90.2 | 72.6 | 47.4 | 64.5 | 74.4 | 62.9 | **77.0** | 84.9 | **92.2** | 94.5 |
| ReLICv2-5k (ours) | 78.3 | 89.9 | 73.0 | **49.4** | **65.6** | **76.9** | **65.5** | 76.8 | 85.1 | 91.4 | 95.7 |

Table 14: Accuracy (in %) of transfer performance of a ResNet50 pretrained on JFT under the linear transfer evaluation protocol. xk refers to the length of pretraining in ImageNet-equivalent epochs, e.g. 1k corresponds to 1000 ImageNet-equivalent epochs of pretraining.

## D.3    ROBUSTNESS AND OOD GENERALIZATION

We also tested the robustness and out-of-distribution (OOD) generalization of RELICv2 representations pretrained on JFT. We use the same set-up described in B.4 where we freeze the pretrained representations on JFT-300M, train a linear classifier using the labelled ImageNet training set and perform zeroshot test evaluation on datasets testing robustness and OOD generalization. As in B.4, we evaluated robustness using the ImageNetV2 [Recht et al., 2019] and ImageNet-C [Hendrycks and Dietterich, 2019] datasets and OOD generalization using ImageNet-R [Hendrycks et al., 2021], ImageNetSketch [Wang et al., 2019] and ObjectNet [Barbu et al., 2019] datasets. We report the robustness results in table 15a and the OOD generalization results in table 15b. We notice that RELICv2 representations pretrained on JFT-300M for different number of ImageNet-equivalent epochs have worse robustness and OOD generalization performance compared to RELICv2 representations pretrained directly on ImageNet (see table 3 for reference). Given that the above datasets have been specifically constructed to measure the robustness and OOD generalization abilities of models pretrained on ImageNet (as they have been constructed in relation to ImageNet), this result is not entirely surprising. We hypothesize that this is due to there being a larger discrepancy between datasets and JFT-300M than these datasets and ImageNet and as such JFT-300M-pretrained representations perform worse than ImageNet-pretrained representations. Additionally, note that pretraining on JFT-300M for longer does not necessarily result in better downstream performance on the robustness and out-of-distribution datasets.

| Epochs | MF | T-0.7 | Ti | IN-C |
|--------|------|-------|------|------|
| 1000 | 57.6 | 66.7 | 73.0 | 32.9 |
| 3000 | 58.6 | 67.5 | 73.4 | 32.8 |
| 5000 | 59.1 | 67.3 | 73.3 | 33.5 |

(a) ImageNetv2 dataset.

| Epochs | IN-R | IN-Sketch | ObjectNet |
|--------|------|-----------|-----------|
| 1000 | 20.4 | 6.7 | 20.3 |
| 3000 | 20.3 | 8.7 | 21.3 |
| 5000 | 20.3 | 5.4 | 20.9 |

(b)  ImageNet-R,  ImageNet-Sketch  and  ObjectNet datasets.

Table 15: Top-1 Accuracy (in %) under linear evaluation on the the ImageNet-R (IN-R), ImageNet-Sketch (IN-S) and ObjectNet out-of-distribution datasets and on ImageNetV2 dataset for RELICv2 pre-trained on JFT-300M for different numbers of ImageNet-equivalent epochs. We evaluate on all three variants on ImageNetV2 – matched frequency (MF), Threshold 0.7 (T-0.7) and Top Images (TI). The results for ImageNet-C (IN-C) are averaged across the 15 different corruptions.

# E  ReLICv2 PSEUDO-CODE IN JAX

Listing 1 provides PyTorch-like pseudo-code for ReLICv2 detailing how we apply the saliency masking and how the different views of data are combined in the target network setting. We also provide a direct comparison with the pseudo-code for ReLIC [Mitrovic et al., 2021] highlighted in listing 2. Note that loss_relic is computed using equation 3.

[H]

```
1  '''
2  f_o: online network: encoder + comparison_net
3  g_t: target network: encoder + comparison_net
4  gamma: target EMA coefficient
5  n_e: number of negatives
6  p_m: mask apply probability
7  '''
8  for x in batch: # load a batch of B samples
9      # Apply saliency mask and remove
        background
10     x_m = remove_background(x)
11     for i in range(num_large_views):
12         # Select either original or
        background-removed
13         # Image with probability p_m
14         x = Bernoulli(p_m) ? x_m : x
15         # Do large random view and augment
16         xl_i = aug(crop_l(x))
17
18         ol_i = f_o(xl_i)
19         tl_i = g_t(xl_i)
20
21     for i in range(num_small_views):
22         # Do small random view and augment
23         xs_i = aug(crop_s(x))
24         # Small views only go through the
        online network
25         os_i = f_o(xs_i)
26
27     loss = 0
28     # Compute loss between all pairs of large
         views
29     for i in range(num_large_views):
30         for j in range(num_large_views):
31             loss += loss_relic(ol_i, tl_j,
        n_e)
32
33     # Compute loss between small views and
         large views
34     for i in range(num_small_views):
35         for j in range(num_large_views):
36             loss += loss_relic(os_i, tl_j,
        n_e)
37     scale = (num_large_views +
        num_small_views) *
38             num_large_views
39     loss /= scale
40
41     # Compute grads, update online and target
         networks
42     loss.backward()
43     update(f_o)
44     g_t = gamma * g_t + (1 - gamma) * f_o
```

Listing 1: Pseudo-code for ReLICv2.

[H]

```
1  '''
2  f_o: online network: encoder + comparison_net
3  g_t: target network: encoder + comparison_net
4  gamma: target EMA coefficient
5  n_e: number of negatives
6  '''
7  # load a batch of B samples
8  for x in batch:
9      # Apply augmentations
10     x1 = aug(x)
11     x2 = aug(x)
12
13     o1, o2 = f_o(x1), f_o(x2)
14     t1, t2 = f_t(x1), f_t(x2)
15
16     # Compute loss between augmented views
17     loss  = loss_relic(o1, t2, n_e) +
18             loss_relic(o2, t1, n_e)
19     loss /= 2
20
21     loss.backward()
22
23     # Compute grads, update online and target
         networks
24     loss.backward()
25     update(f_o)
26     g_t = gamma * g_t + (1 - gamma) * f_o
```

Listing 2: Pseudo-code for ReLIC.

## F    IMAGE PREPROCESSING

### F.1    AUGMENTATIONS

Following the data augmentations protocols of [Chen et al., 2020a; Grill et al., 2020; Caron et al., 2020], ReLICv2 uses a set of augmentations to generate different views of the original image which has three channels, red $r$, green $g$ and blue $b$ with $r, g, b \in [0, 1]$.

The augmentations used, in particular (corresponding to `aug` in Listing 2) are the same as in [Grill et al., 2020] and are generated as follows; for exact augmentations parameters see table 16). The following sequence of operations is performed in the given order.

1. Crop the image: Randomly select a patch of the image, between a minimum and maximum crop area of the image, with aspect ratio sampled log-uniformly in $[3/4, 4/3]$. Upscale the patch, via bicubic interpolation, to a square image of size $s \times s$.

2. Flip the image horizontally.

3. Colour jitter: randomly adjust brightness, contrast, saturation and hue of the image, in a random order, uniformly by a value in $[-a, a]$ where $a$ is the maximum adjustment (specified below).

4. Grayscale the image, such that the channels are combined into one channel with value $0.2989r + 0.5870g + 0.1140b$.

5. Randomly blur. Apply a $23 \times 23$ Gaussian kernel with standard deviation sampled uniformly in $[0.1, 2.0]$.

6. Randomly solarize: threshold each channel value such that all values less than $0.5$ are replaced by $0$ and all values above or equal to $0.5$ are replaced with $1$.

Apart from the initial step of image cropping, each step is executed with some probability to generate the final augmented image. These probabilities and other parameters are given in table 16, separately for augmenting the original image $x_i$ and the positives $\mathcal{P}(x_i)$. Note that we use 4 large views of size $224 \times 224$ pixels and 2 small views of $96 \times 96$ pixels; to get the first and third large views and the first small view we use the parameters listed below for odd views, while for the second and fourth large view and the second small view we use the parameters for even views.

| Parameter | Even views | Odd views |
|---|---|---|
| Probability of randomly cropping | 50% | 50% |
| Probability of horizontal flip | 50% | 50% |
| Probability of colour jittering | 80% | 80% |
| Probability of grayscaling | 20% | 20% |
| Probability of blurring | 100% | 10% |
| Probability of solarization | 0% | 20% |
| Maximum adjustment $a$ of brightness | 0.4 | 0.4 |
| Maximum adjustment $a$ of contrast | 0.4 | 0.4 |
| Maximum adjustment $a$ of saturation | 0.2 | 0.2 |
| Maximum adjustment $a$ of hue | 0.1 | 0.1 |
| Crop size $s$ | 224 | 96 (small), 224 (large) |
| Crop minimum area | 8% | 5% (small), 14% (large) |
| Crop maximum area | 100% | 14% (small), 100% (large) |

Table 16: Parameters of data augmentation scheme. Small/large indicates small or large crop.

### F.2    SALIENCY MASKING

Using unsupervised saliency masking enables us to create positives for the anchor image with the background largely removed and thus the learning process will rely less on the background to form representations. This encourages the representation to localize the objects in the image [Zhao et al., 2021].

We develop a fully unsupervised saliency estimation method that uses the self-supervised refinement mechanism from DeepUSPS [Nguyen et al., 2019] to compute saliency masks for each image in the ImageNet training set. By applying the saliency masks on top of the large views, we obtain masked images with the background removed. To further increase the background variability, instead of using a black background for the images, we apply a homogeneous grayscale to the background with the grayscale level randomly sampled for each image during training. We also use a foreground threshold such that we apply the saliency mask only if it covers at least $5\%$ of the image. The masked images with the grayscaled background are used only during training. Specifically, with a small probability $p_m$ we selected the masked image of the large view in place of the large view. Figure 8 shows how the saliency masks are added on top of the images to obtain the images with grayscale background.

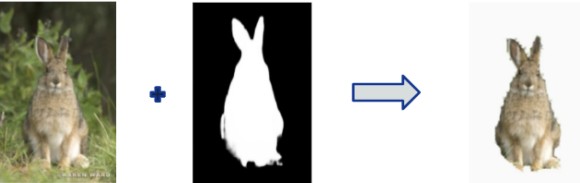

Figure 8: Illustration of how for each image in the ImageNet training set (left) we use our unsupervised version of DeepUSPS to obtain the saliency mask (middle) which we then apply on top of the image to obtain the image with the background removed (right).

### F.2.1 Training the saliency detection network to obtain saliency masks

DeepUSPS [Nguyen et al., 2019] is a saliency prediction method that uses self-supervision to refine pseudo-labels from a number of handcrafted saliency methods. To obtain saliency masks for the images in ImageNet, we build a new saliency detection method that leverages the self-supervised refinement mechanism from DeepUSPS [Nguyen et al., 2019]. To this end, we firstly sample a random subset of 2500 ImageNet images; note that the original implementation of DeepUSPS uses 2500 images from the MSRA-B dataset. We instead use a randomly selected subset of the ImageNet training set of the same size to ensure a fair comparison to previous work. We compute initial saliency masks for the 2500 ImageNet images using the following handcrafted methods: Robust Background Detection (RBD) [Zhu et al., 2014], Manifold Ranking (MR) [Yang et al., 2013], Dense and Sparse Reconstruction (DSR) [Li et al., 2013] and Markov Chain (MC) [Jiang et al., 2013]. Note that these methods do not make use of any supervised label information.

We then follow the two-stage mechanism proposed by DeepUSPS [Nguyen et al., 2019] to obtain a saliency prediction network. In the first stage, the noisy pseudo-labels from each handcrafted method are iteratively refined. In the second stage, these refined labels from each handcrafted saliency method are used to train the final saliency detection network. The saliency detection network is then used to compute the saliency masks for all images in the ImageNet training set. For the refinement procedure and for training the saliency detection network, we adapt the publicly available code for training DeepUSPS: `https://tinyurl.com/wtlhgo3`. Note that the official implementation for DeepUSPS uses as backbone a DRN-network [Yu et al., 2017] which was pretrained on CityScapes [Cordts et al., 2016] with supervised labels. To be consistent with our fully-unsupervised setting, we replace this network with a ResNet50 2x model which was pretrained on ImageNet using the self-supervised objective from SWaV [Caron et al., 2020]. We used the publicly available pretrained SWaV model from: `https://github.com/facebookresearch/swav`.

To account for this change in the architecture, we adjust some of the hyperparameters needed for the the two-stage mechanism of DeepUSPS. In the first stage, the pseudo-generation networks used for refining the noisy pseudo-labels from each of the handcrafted methods are trained for 25 epochs in three self-supervised iterations. We start with a learning rate of $1e-5$ which is doubled during each iteration. In the second stage, the saliency detection network is trained for 200 epochs using a learning rate of $1e-5$. We use the Adam optimizer with momentum set to 0.9 and a batch size of 10. The remaining hyperparameters are set in the same way as they are in the original DeepUSPS code.

Next, for different probabilities $p_m$ of removing the background of the large augmented views during training, we report below the top-1 accuracy under linear evaluation on ImageNet.

| $p_m$ | 0.0 | 0.1 | 0.15 | 0.2 | 0.25 |
|-------|-----|-----|------|-----|------|
| Top-1 | 76.8 | **77.1** | 76.8 | 76.8 | 76.7 |

Applying the saliency masks 10% of the time results in the best performance and significantly improves over not using masking ($p_m = 0$).

### F.2.2 EXAMPLES OF SALIENCY MASKS

In Figure 9 we provide a random sample of the saliency masks obtained from the saliency detection network that are used as part of the data augmentations pipeline in RELICv2 pretraining. We notice that while the obtained saliency masks can, in general, segment the object from the background well, some common failure cases include the setting where the object fills the entire image (i.e. there is no background), or when the object is difficult to distinguish from the background. This is why, we believe it is important to only use the saliency masking with a small probability during training.

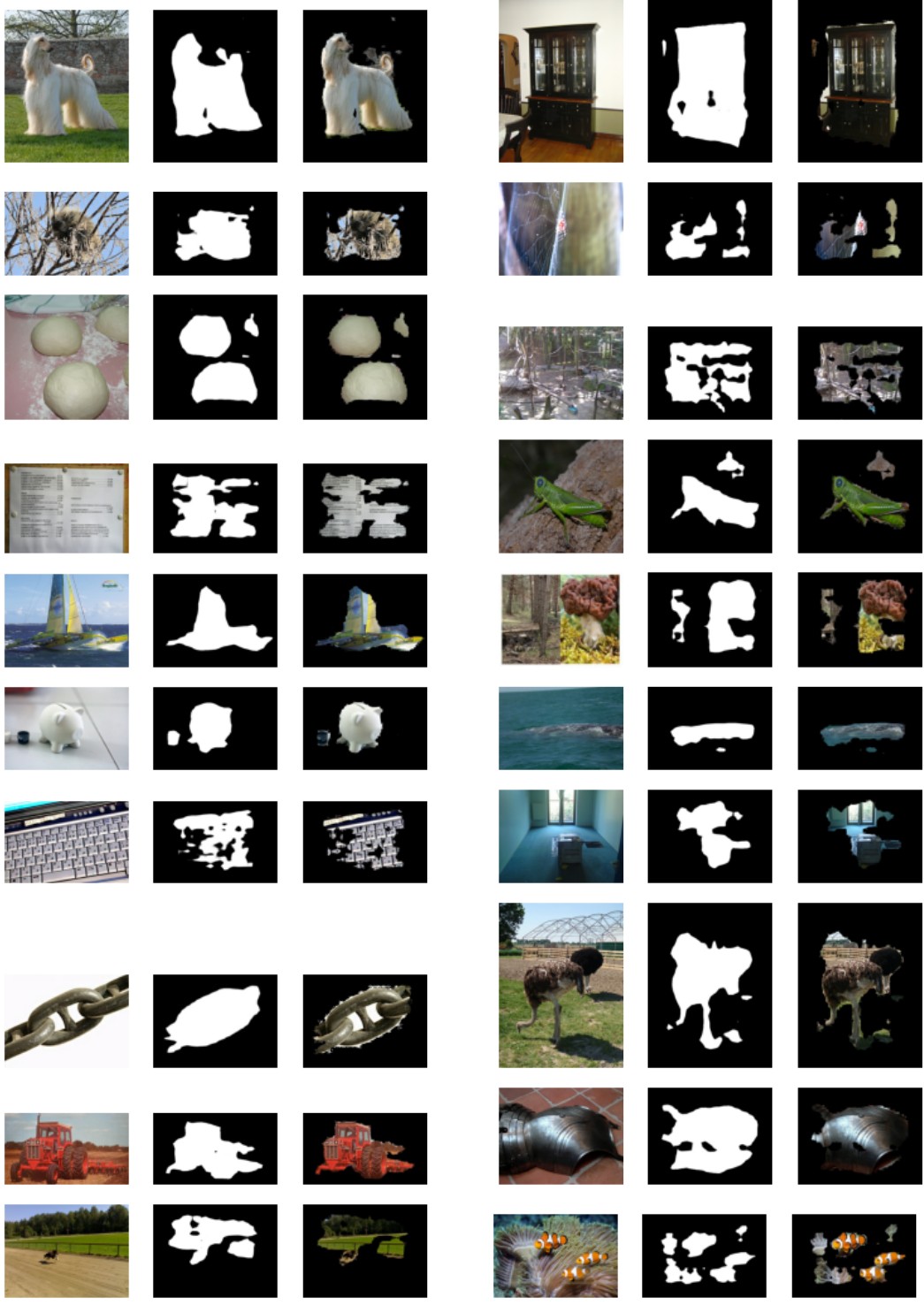

Figure 9: Examples of saliency masks used during training ReLICv2. Each of the two columns contains randomly selected examples of image (left), mask obtained from our saliency detection networks (middle) and image with applied masks, without background grayscaling (right).

# G    ABLATIONS

In order to determine the sensitivity of RELICv2 to different model hyperparameters, we perform an extensive ablation study. Unless otherwise noted, in this section we report results after 300 epochs of pretraining. As saliency masking is one of the main additions of RELICv2 on top of RELIC and was not covered extensively in the main text, we start our ablation analysis with looking into the effect of different modelling choices for it.

## G.1    USING DIFFERENT DATASETS FOR OBTAINING THE SALIENCY MASKS

In the main text in Sections 3, 4.2, 4.1 we used a saliency detection network trained only on a randomly selected subset of 2500 ImageNet images using the refinement mechanism proposed by DeepUSPS [Nguyen et al., 2019]. Here we explore whether using additional data could help improve the performance of the saliency estimation and of the overall representations learnt by RELICv2. For this purpose, we use the MSRA-B dataset [Liu et al., 2010], which was originally used by DeepUSPS to train their saliency detection network. MSRA-B consists of 2500 training images for which handcrafted masks computed with the methods Robust Background Detection (RBD) [Zhu et al., 2014], Hierarchy-associated Rich Features (HS) [Zou and Komodakis, 2015], Dense and Sparse Reconstruction (DSR) [Li et al., 2013] and Markov Chain (MC) [Jiang et al., 2013] are already available. We use the same hyperparameters as described in Section F.2.1 to train our saliency detection network on MSRA-B; note that these are the same hyperparameters we use for training our saliency detection network on ImageNet.

We explored whether using saliency masks obtained from training the saliency detection network on the MSRA-B affects performance of RELICv2 pre-training on ImageNet. We noticed that for RELICv2 representations pretrained on ImageNet for 1000 epochs, we get 77.2% top-1 and 93.3% top-5 accuracy under linear evaluation on the ImageNet validation set for a ResNet50 (1x) encoder. The slight performance gains may due to the larger variety of images in MSRA-B used for training the saliency detection network, as opposed to the random sample of 2500 ImageNet images that we used for training the saliency detection network directly on the ImageNet dataset.

We also explored training the saliency detection network on 5000 randomly selected images from the ImageNet dataset and this resulted in the model overfitting, which degraded the quality of the saliency masks and resulted in a RELICv2 performance of 76.7% top-1 and 93.3% top-5 accuracy on the ImageNet validation set after 1000 epochs of pretraining on ImageNet training set.

The results for RELICv2 in Section D are obtained by applying the saliency detection network trained on MSRA-B to all images in JFT-300M and then applying the saliency masks to the large augmented views during training as described in Section F.2.

## G.2    ANALYSIS AND ABLATIONS FOR SALIENCY MASKS

Using saliency masking during RELICv2 training enables us to learn representations that focus on the semantically-relevant parts of the image, i.e. the foreground objects, and as such the learned representations should be more robust to background changes. We investigate the impact of using saliency masks with competing self-supervised benchmarks, the effect of the probability $p_m$ of applying the saliency mask to each large augmented view during training as well as the robustness of RELICv2 to random masks and mask corruptions. For the ablation experiments described in this section, we train the models for 300 epochs.

**Using saliency masks with competing self-supervied methods.**    We evaluate the impact of using saliency masks with competing self-supervised methods such as BYOL [Grill et al., 2020]. This method only uses two large augmentented views during training and we randomly apply the saliency masks, in a similar way as described in Section F.2, to each large augmented view with probability $p_m$. We report in table 17 the top-1 and top-5 accuracy under linear evaluation on ImageNet for different settings of $p_m$ for removing the background of the augmented images. We notice that saliency masking also helps to improve performance of BYOL.

**Mask apply probability.**    We also investigate the effect of using probabilities ranging from 0 to 1 for applying the saliency mask during training for RELICv2. In addition, we explore further the

| | Mask probability $p_m$ | 0 | 0.1 | 0.15 | 0.2 | 0.25 | 0.3 |
|---|---|---|---|---|---|---|---|
| BYOL | Top-1 | 73.1 | 73.4 | 73.2 | 73.3 | 72.8 | 71.8 |
| | Top-5 | 91.2 | 91.3 | 91.2 | 91.3 | 90.8 | 90.1 |

Table 17: Top-1 and top-5 accuracy (in %) under linear evaluation on the ImageNet validation set for BYOL trained using different probabilities of using the saliency mask to remove the background of the augmented images. Models are trained for 300 epochs.

effect of using different datasets for training the saliency detection network that is subsequently used for computing the saliency masks. Table 18 reports the top-1 and top-5 accuracy for varying the mask apply probability $p_m$ between 0 and 1 and for using the ImageNet vs. the MSRA-B dataset [Liu et al., 2010] for training our saliency detection network. Note that using the additional images from the MSRA-B dataset to train the saliency detection network results in better saliency masks which translates to better performance when using the saliency masks during RELICv2 training.

| Mask probability $p_m$ | Sal. network trained on ImageNet | | Sal. network trained on MSRA-B | |
|---|---|---|---|---|
| | Top-1 | Top-5 | Top-1 | Top-5 |
| 0 | 75.2 | 92.4 | 75.2 | 92.4 |
| 0.05 | 75.3 | 92.6 | 75.2 | 92.6 |
| 0.1 | 75.4 | 92.5 | 75.3 | 92.4 |
| 0.15 | 75.2 | 92.5 | 75.5 | 92.5 |
| 0.2 | 75.2 | 92.5 | 75.6 | 92.6 |
| 0.25 | 75.0 | 92.3 | 75.3 | 92.5 |
| 0.3 | 75.1 | 92.3 | 74.8 | 92.4 |
| 0.4 | 75.0 | 92.3 | 75.3 | 92.5 |
| 0.5 | 74.7 | 92.2 | 75.0 | 92.4 |
| 0.6 | 75.0 | 92.3 | 75.0 | 92.3 |
| 0.7 | 74.4 | 92.3 | 74.6 | 92.0 |
| 0.8 | 73.9 | 91.7 | 75.0 | 92.1 |
| 0.9 | 74.0 | 91.7 | 74.6 | 92.0 |
| 1.0 | 73.7 | 91.7 | 74.5 | 92.0 |

Table 18: Top-1 and top-5 accuracy (in %) under linear evaluation on the ImageNet validation set for a ResNet50 (1x) encoder set for different probabilities $p_m$ of using the saliency mask to remove the background of the large augmented views during training and for using different datasets to train the saliency detection network for computing the saliency masks. Models are trained for 300 epochs.

**Random masks and mask corruptions.** To understand how important having accurate saliency masks for the downstream performance of representations is we also investigated using random masks, corrupting the saliency masks obtained from our saliency detection network and using a bounding box around the saliency masks during RELICv2 training.

We explored using completely random masks, setting the saliency mask to be a random rectangle of the image and also a centered rectangle. As ImageNet images generally consists of images with objects centered in the middle of the image, we expect that using a random rectangle that is centered around the middle will cover a reasonable portion of the object. Table 19 reports the performance under linear evaluation on the ImageNet validation set when varying the size of the random masks to cover different percentage areas $a_p$ of the full image. We notice that improving the quality of the masks, by using random rectangle patches instead of completely random points in the image as the mask, results in better performance. However, the performance with random masks is $> 1\%$ lower than using saliency masks from our saliency detection network. As expected, using centered rectangles instead of randomly positioned rectangles as masks results in better peformance.

Moreover, to test the robustness of RELICv2 to corruptions of the saliency masks, we add/remove from the masks a rectangle proportional to the area of the saliency mask. The mask rectangle is added/removed from the image center. Table 20 reports the results when varying the area of the rectangle to be added/removed to cover different percentages $m_p$ of the saliency masks. We notice

| Image percentage area $a_p$ | Random | | Rectangle | | Centered Rectangle | |
|---|---|---|---|---|---|---|
| | Top-1 | Top-5 | Top-1 | Top-5 | Top-1 | Top-5 |
| 10% | 70.8 | 89.9 | 70.9 | 90.3 | 71.3 | 90.1 |
| 20% | 72.2 | 90.7 | 73.1 | 91.3 | 73.4 | 91.3 |
| 30% | 72.9 | 91.3 | 73.8 | 91.8 | 73.8 | 91.9 |
| 40% | 73.1 | 91.4 | 74.2 | 91.9 | 74.1 | 92.0 |
| 50% | 73.3 | 91.5 | 74.0 | 92.0 | 74.3 | 92.0 |
| 60% | 73.6 | 91.8 | 74.2 | 92.1 | 74.3 | 92.2 |
| 70% | 73.7 | 91.9 | 74.4 | 92.1 | 74.4 | 92.2 |
| 80% | 74.1 | 92.1 | 74.4 | 92.2 | 74.2 | 92.1 |
| 90% | 74.1 | 92.2 | 74.4 | 92.1 | 74.2 | 92.2 |

Table 19: Top-1 and top-5 accuracy (in %) under linear evaluation on the ImageNet validation set for a ResNet50 (1x) encoder set for using different types of random masks that cover various percentage areas ($a_p$) of the full image. These random masks are applied on top of the large augmented views during training with probability 0.1. Models are trained for 300 epochs.

that while RELICv2 is robust to small corruptions of the saliency mask its performance drops in line with the quality of the saliency masks degrading.

| Mask percentage area $m_p$ | Add rectangle to mask | | Remove rectangle from mask | |
|---|---|---|---|---|
| | Top-1 | Top-5 | Top-1 | Top-5 |
| 10% | 75.2 | 92.5 | 75.2 | 92.3 |
| 20% | 75.3 | 92.6 | 75.1 | 92.4 |
| 30% | 75.1 | 92.3 | 74.7 | 92.2 |
| 40% | 74.9 | 92.2 | 74.6 | 92.2 |
| 50% | 74.9 | 92.4 | 74.5 | 92.0 |
| 60% | 74.9 | 92.2 | 74.0 | 91.7 |
| 70% | 74.8 | 92.2 | 73.6 | 91.7 |
| 80% | 74.8 | 92.4 | 73.4 | 91.4 |
| 90% | 74.7 | 92.2 | 73.0 | 91.3 |
| 100% | 74.6 | 92.3 | 72.6 | 90.9 |

Table 20: Top-1 and top-5 accuracy (in %) under linear evaluation on the ImageNet validation set for a ResNet50 (1x) encoder set for corrupting the saliency masks by adding/remove a rectangle from the image center. The rectangle is a percentage ($m_p$) of the saliency mask area (the higher the percentage the higher the corruption). The corrupted saliency masks are applied on top of the large augmented views during training with probability 0.1.

Finally, we also explore corrupting the masks using a bounding box around the saliency mask which results in 74.5% top-1 and 92.2% top-5 accuracy under linear evaluation on the ImageNet validation set for a ResNet50 (1x) encoder trained for 300 epochs with mask apply probability of 0.1 Note that this performance is comparable to using random rectangles to mask the large augmented views during training (see table 19) and is lower than directly using the saliency masks from the trained saliency detection network.

### G.3 BENEFITS OF SALIENCY MASKING FOR ROBUSTNESS AND OOD GENERALIZATION

To further highlight the benefits of using saliency masking, we conducted an additional ablation where we show the impact of only using the multiple views on downstream performance, particularly on robustness and OOD generalization downstream tasks. In Table 21, we compare the performance of ReLICv2 without saliency masking (just using multi-crop) vs ReLICv2 with saliency masking. We report Top-1 Accuracy under linear evaluation on ImageNetV2 and ImageNet-C (robustness), and ImageNet-R (IN-R), ImageNet-Sketch (IN-S) and ObjectNet (out-of-distribution). ImageNetv2 has three variants – matched frequency (MF), Threshold 0.7 (T-0.7) and Top Images (TI). The results for ImageNet-C (IN-C) are averaged across the 15 different corruptions. These results highlight that the use of saliency masking for ReLICv2 results in more robust representations.

Table 21: Top-1 Accuracy (in %) under linear evaluation on ImageNetV2 and ImageNet-C (robustness), and ImageNet-R (IN-R), ImageNet-Sketch (IN-S) and ObjectNet (out-of-distribution). ImageNetv2 has three variants – matched frequency (MF), Threshold 0.7 (T-0.7) and Top Images (TI). The results for ImageNet-C (IN-C) are averaged across the 15 different corruptions.

| | | Robustness | | | OOD Generalization | | |
|---|---|---|---|---|---|---|---|
| Method | MF | T-0.7 | Ti | IN-C | IN-R | IN-S | ObjectNet |
| RELICv2 w/o saliency masking | 64.2 | 73.1 | 78.0 | 43.8 | 22.9 | 5.7 | 24.5 |
| RELICv2 (ours) | 65.3 | 74.5 | 79.4 | 44.8 | 23.9 | 9.9 | 25.9 |

## G.4 OTHER MODEL HYPERPARAMETERS

Now we turn our attention to ablating the effect of other model hyperparameters on the downstream performance of RELICv2 representations. Note that these hyperparameters have been introduced and extensively ablated in prior work [Grill et al., 2020; Mitrovic et al., 2021; 2020].

**Number of negatives.** As mentioned in Section 2 RELICv2 selects negatives by randomly subsampling the minibatch in order to avoid false negatives. We investigate the effect of changing number of negatives in table 22. We can see that the best performance can be achieved with relatively low numbers of negatives, i.e. just 10 negatives. Furthermore, we see that using the whole batch as negatives has one of the lowest performances.

In further experiments, we observed that for longer pretraining (e.g. 1000 epochs) there is less variation in performance than for pretraining for 300 epoch which itself is also quite low.

| Number of negatives | Top-1 | Top-5 |
|---|---|---|
| 1 | 75.1 | 92.4 |
| 5 | 75.2 | 92.6 |
| 10 | 75.4 | 92.5 |
| 20 | 75.3 | 92.7 |
| 50 | 75.5 | 92.5 |
| 100 | 75.4 | 92.5 |
| 500 | 75.1 | 92.4 |
| 1000 | 75.3 | 92.6 |
| 2000 | 75.4 | 92.5 |
| 4096 | 75.2 | 92.6 |

Table 22: Top-1 and top-5 accuracy (in %) under linear evaluation on the ImageNet validation set for a ResNet50 (1x) encoder set for different numbers of randomly selected negatives. All settings are trained for 300 epochs.

**Target EMA.** RELICv2 uses a target network whose weights are an exponential moving average (EMA) of the online encoder network which is trained normally using stochastic gradient descent; this is a setup first introduced in [Grill et al., 2020] and subsequently used in [Mitrovic et al., 2021] among others. The target network weights at iteration $t$ are $\xi_t = \gamma \xi_{t-1} + (1 - \gamma)\theta_t$ where $\gamma$ is the EMA parameter which controls the stability of the target network ($\gamma = 0$ sets $\xi_t = \theta_t$); $\theta_t$ are the parameters of the online encoder at time $t$, while $\xi_t$ are the parameters of the target encoder at time $t$. As can be seen from table 23, all decay rates between 0.9 and 0.996 yield similar performance for top-1 accuracy on the ImageNet validation set after pretraining for 300 epochs indicating that RELICv2 is robust to choice of $\gamma$ in that range. For values of $\gamma$ of 0.999 and higher, the performance quickly degrades indicating that the updating of the target network is too slow. Note that contrary to [Grill et al., 2020] where top-1 accuracy drops below 20% for $\gamma = 1$, RELICv2 is significantly more robust to this setting achieving double that accuracy.

| $\gamma$ | Top-1 | Top-5 |
|---|---|---|
| 0 | 73.5 | 91.5 |
| 0.9 | 74.6 | 92.2 |
| 0.99 | 75.5 | 92.6 |
| 0.993 | 75.4 | 92.5 |
| 0.996 | 74.4 | 92.0 |
| 0.999 | 70.5 | 89.8 |
| 1.0 | 39.6 | 63.6 |

Table 23: Top-1 and top-5 accuracy (in %) under linear evaluation on the ImageNet validation set for a ResNet50 (1x) encoder set for different setting of the target exponentially moving average (EMA). All settings are trained for 300 epochs.

