# OpenReview forum: "Pushing the limits of self-supervised learning: Can we outperform supervised learning without labels?"
_ICLR.cc/2023/Conference — Submitted to ICLR 2023_

### Official Review · Reviewer_hHxH · 2022-10-18

**Confidence:** 4
**Correctness:** 4
**Technical Novelty And Significance:** 2
**Empirical Novelty And Significance:** 2
**Recommendation:** 5

**Clarity, Quality, Novelty And Reproducibility:**

Clarity: paper was carefully written with good clarity
Quality: extensive experimental results, though without uncertainties reporting
Novelty: The main contribution is additional augmentation method
Reproducibility: details are given, not sure if models are easy to tune

**Strength And Weaknesses:**

(1) Strength: a lot of experiments are being carried out with a lot of analysis
(2) Strength: Results seems to be better than previous methods.
(3) Weakness: However there is no error bars to convince readers about the robustness of the reported results. for example, multiple duplicated experiments should be perform and p-value calculated when comparing the results of various methods.
(4) Weakness: it is not clear if the reported results are due to more careful tuning of hyper-parameters. A good way to silence this criticism is to show evidence of robustness of results with respect to hyper-parameter values.
(5) Weakness: Seems that the major contributions are two more augmentation methods.

**Summary Of The Paper:**

This paper is an extension of RELIC self supervised learning method.

In this extension, two main augmentation method is added. (1) Saliency map (2) Views of varying sizes.
The loss function has also been modified.

Extensive experiments had been perform to study the effects of these extensions.

**Summary Of The Review:**

Well written paper, however paper can be strengthen with error bars and with experiments showing how results differs with different hyper-parameters.

---

> ### Author Response · Authors · 2022-11-11
> **Reviewer hHxH**
>
> We thank the reviewer for their time and effort in reviewing our paper and for their insightful comments and suggestions on how to improve the paper.
>
> Please allow us to clarify below the reviewer’s concerns.
>
> ### Uncertainties
>
> Thank you for suggesting to incorporate error bars to be able to better quantify the robustness of the reported results. While this practice is not standard in the self-supervised literature (and the reason we didn’t include it in our work initially), we have started experiments to evaluate the variance in our results and will update this thread with the results as soon as possible.
>
> ### Hyper-parameter tuning
>
> We completely agree that it is important to understand the nature of our reported improvements, both in terms of ablation, as well as with regards to baselines, comparisons, and hyperparameter tuning. With respect to baselines, we would like to clarify that we take the reported best results of those baselines from their respective papers, rather than re-training the models ourselves. For ReLICv2, we use all the same hyperparameters as in the original ReLIC paper apart from the new additions of multiple views and saliency masking. We follow our proposed design choice of having a large number of large views and a small number of small views and ablate the exact choices in Section 3.1. For saliency masking, the probability of applying the masking is a hyperparameter and we ablate this choice in Appendix F.2., where it can be seen that ReLICv2 is reasonably robust to a range of similar saliency masking probabilities.
>
> In summary, the performance gain seen in ReLICv2 comes from improvements in the method and not hyperparameter tuning.
>
> ### Two more augmentation methods
>
> In this work, we propose to add two inductive biases (multiple views and saliency masking) to self-supervised learning methods. By incorporating these inductive biases we develop the self-supervised method ReLICv2 and conduct an extensive experimental evaluation showing significant improvements in performance across a very wide range of diverse datasets and tasks, as well as pre-training datasets. While these two inductive biases can be viewed as simple additions, they bring significant performance gains and are general (as also highlighted by Reviewer jUzd). In particular, they can be applied together with other self-supervised methods (e.g. SimCLR, BYOL) as well as in conjunction with other architectures such as Vision Transformers. Furthermore, note that adding these two inductive biases enables us, for the first time in the literature, to learn representations without labels that outperform supervised learning baselines across a wide range of ResNet encoder architectures.
>
> Thus, we believe that our work which is general, simple and performant represents a valuable contribution to self-supervised representation learning and can be of significant interest to the machine learning community in general.
>
> ### Reproducibility
>
> For transparency and ease of reproducibility, we have provided extensive explanations of the experimental settings in the appendix that should enable the complete reimplementation and reproduction of all the experiments in this paper. We are also working on open-sourcing pretrained models and the code for our method on Github which will further enable the ease of reproducibility; we hope to provide the pretrained models and code as soon as possible.

---

> ### Author Response · Authors · 2022-11-16
> **Follow-up**
>
> Dear reviewer hHxH,
>
> Thank you once again for your valuable feedback! We are sincerely grateful for your time and energy in the review process. We hope that our responses have addressed your concerns. If you have any remaining concerns, please let us know. We will do our utmost to address them.
>
> Thank you!
>
> Paper5023 Authors

---

> ### Author Response · Authors · 2022-11-18
> **Additional results**
>
> We thank the reviewer again for their time and thoughtful comments which have helped us to improve our paper.
> - To highlight the generality of the two proposed inductive biases (multiple views and saliency masking), we combined these inductive biases with another performant self-supervised objective BYOL; we choose BYOL as it is a non-contrastive method and allows us to highlight the generality of the proposed inductive biases for self-supervised learning. We use 4 large views and 2 small views (as we also use in ReLICv2 to ensure a fair comparison) and use the same saliency masking as in ReLICv2, again to ensure a fair comparison. The original BYOL method achieves 74.3% top-1 accuracy on ImageNet, while BYOL with multiple views and saliency masking achieves 75.2%, an improvement of almost 1%. Moreover, please note that in 3.1, we already provide results to assess the importance of enforcing invariance over background removal and object styles (as encoded through multiple views and saliency masking) and compare ReLICv2 to SimCLR with multiple views and saliency masking when pre-training on 100 epochs.
> - We also performed an analysis on the variability of learning representations with ReLICv2 as measured by top-1 ImageNet accuracy and calculated a standard error of 0.09% across 5 runs.

---

> ### Author Response · Authors · 2022-12-06
> **Dear Reviewer hHxH**
>
> Thank you again for your time, commitment, constructive comments and interesting questions during the review process!
>
> We hope our responses (Nov 11 & Nov 18) have addressed in detail the raised questions and comments. We have correspondingly updated the paper (Nov 18) and have furthermore released checkpoints and evaluation code for ReLICv2 (Nov 25).
>
> Please let us know if there are any outstanding questions or comments. We would be very happy to address these! :)
>
> If we have satisfactorily addressed the raised questions and concerns, we would kindly ask you to reevaluate your evaluation in light of this and consider raising your score.
>
> Thank you very much for your consideration of our paper!
>
> Paper5023 Authors

---

### Official Review · Reviewer_Z39N · 2022-10-23

**Confidence:** 4
**Correctness:** 4
**Technical Novelty And Significance:** 2
**Empirical Novelty And Significance:** 3
**Recommendation:** 6

**Clarity, Quality, Novelty And Reproducibility:**

The paper is well written.
All experimental details are provided.

**Strength And Weaknesses:**

Strengths:
1. This method achieves state-of-the-art ImageNet linear probe accuracy. It combines a few existing effective building components including ReLIC loss function, multicrops and background mask. Even though these components are already existing in the literature, I find that some special settings including 4x big + 2x small multicrops are novel.
2. The authors conducted a scaling study on a larger dataset such as JFT300M. Even though they derived relatively poor scaling performance, I find such study helpful and insightful, i.e. how augmentation-based joint-embedding method can benefit from scaling.

Weaknesses:
1. Overall the paper has very limited novelty.
2. From ablation study, the background mask only shows 0.3% improvements. The main advantage seems to come from the multicrops. Using 4x big views + 2x small views means effectively much higher computation cost. This method still uses 4096 batch size. As far as I can tell, the previous works limiting to 2x big views + many small views with this batch size are mainly caused by GPU memory. It is not discussed how their method can afford this.
3. The background mask here is very hand-crafted. Figure 8 seems too good to be true for arbitrary input. Is there any failure case for the background removing?

**Summary Of The Paper:**

This paper proposes a novel self-supervised learning framework called ReLIC v2. This framework builds upon ReLIC's loss and combines multicrops and background mask. The method achieves the state-of-the-art and better linear probe accuracy than supervised ResNet50.
Extensive transfer experiments and ablation study verifies the effectiveness of the proposed method.

**Summary Of The Review:**

State-of-the-art self-supervised learning framework using ReLIC loss plus multicrops and background mask. Overall limited novelty but a lot of novel engineering design that's useful for the community.

---

> ### Author Response · Authors · 2022-11-11
> **Reply for reviewer Z39N [Part 2/2]**
>
> ### Batch size with multicrop - GPU memory
>
> While using multiple crops increases the memory required, we still use the same amount of GPU for 4 large and 2 small views (our default for ReLICv2) as we do for just 2 large views (standard setting in the literature).
>
> ### Examples of saliency masks
>
> Thank you very much for your comment! Our intention with Figure 8 was to show how the saliency masking is applied and not to highlight how good the saliency masks are in general. In fact, it is often the case that saliency masks do not capture all parts of the objects or capture parts of the background. We will add in Appendix F.2.2 a random sample of the computed saliency masks to indicate failure cases. These can also be viewed here: https://ibb.co/1RCcWy5. Each of the two columns in the image contains randomly selected examples of an image (left), the mask obtained from our saliency detection networks (middle) and the image with the applied mask, without background grayscaling (right). We notice that while the obtained saliency masks can, in general, segment the object from the background well, some common failure cases include the setting where the object fills the entire image (i.e. there is no background), or when the object is difficult to distinguish from the background. This is why we believe it is important to only use saliency masking with a small probability during training.
> Moreover, please note that in Appendix G.2 we conduct several ablations to test the importance of the accuracy of the saliency masks for downstream performance. In particular, we explore the effect of using random masks, corrupting the saliency masks obtained from our saliency detection network and using a bounding box around the saliency masks during RELICv2 training.

---

> > ### Comment · Reviewer_Z39N · 2022-12-10
> > **Minor question**
> >
> > Dear authors,
> >
> > Thanks for the detailed response.
> > I do have an extra question on the claim of "beating supervised baseline". In SSL literature, people usually use 76.5% ResNet-50 ImageNet accuracy as the baseline. This number originally comes from Pytorch's official implementation. However, nowadays with better augmentation/training recipe, ResNet-50 ImageNet accuracy has reached almost 80%. I am wondering if you still think this is a fair claim that the SSL method beats the supervised baseline.

---

> > > ### Author Response · Authors · 2022-12-12
> > > **Response**
> > >
> > > Dear Reviewer,
> > >
> > > Thank you very much for your reply! As you pointed out (and we discussed at the beginning of section 3), 76.5% is used as the supervised baseline in the SSL literature. This is the supervised performance under the augmentations introduced in the SimCLR paper. Supervised performances above this number are predominantly achieved by using additional complex and sophisticated augmentation strategies and a number of intricate optimization tricks like casting parameters between different levels of precision. In our work we focus on a like-for-like comparison of self-supervised and supervised learning. To achieve this we use the same architectures and data augmentations for both the supervised and ReLICv2 models in order to ensure a fair comparison. Note that we only add saliency masking on top of the SimCLR augmentations and it has been shown in the literature that saliency masking has a detrimental effect on supervised performance (we also discuss this in section 3). At the beginning of section 3 we comment on the existence of improved supervised models based on improved augmentation and optimization approaches. We believe that these additional tricks can be easily applied within self-supervised methods and we expect them to lead to analogous improvements as in supervised learning.
> > >
> > > We would also like to draw the attention of the reviewer to the fact that ReLICv2 achieves state-of-the-art performance over competing self-supervised methods across a wide variety of architectures and learning settings (ImageNet evaluation, transfer, robustness and ood generalization). We strongly believe that our technical contributions, extensive experiments, state-of-the-art performance and open-sourced checkpoints are a valuable contribution and can be of significant interest to the machine learning community.
> > >
> > > We understand how the statement of beating supervised learning can be viewed as an overclaim. We will further clarify the main contributions of the paper in the manuscript and add more context around the comparison to supervised learning upon acceptance.

---

> ### Author Response · Authors · 2022-11-11
> **Reply for reviewer Z39N [Part 1/2]**
>
> We thank the reviewer for their time and effort in reviewing our paper and for their insightful comments and suggestions on how to improve the paper.
>
> Please allow us to clarify below the reviewer’s concerns.
>
> ### Novelty
>
> We thank the reviewer for their comments and for pointing out the areas we may need to clarify in the draft.
>
> In our paper we propose to add two inductive biases (multiple views and saliency masking) to self-supervised learning. Leveraging these inductive biases we develop the self-supervised method ReLICv2 and conduct an extensive experimental evaluation showing significant improvements in performance across a very wide range of diverse datasets and tasks, as well as pre-training datasets.
>
> While our proposed addition of saliency masking and multiple views can be viewed as simple additions, they result in significant performance improvements and are of a general nature (as also highlighted by Reviewer jUzd). In particular, they can be applied together with other self-supervised methods (e.g. SimCLR, BYOL) as well as in conjunction with other architectures such as Vision Transformers. Furthermore, note that adding these two inductive biases enables us, for the first time in the literature, to learn representations without labels that outperform supervised learning baselines across a wide range of ResNet encoder architectures.
>
> We believe that our work is general, simple and performant, and represents a valuable contribution to self-supervised literature and can be of significant interest to the machine learning community in general.
>
> ### Improvements from background masking
>
> It is indeed the case that for evaluation on the ImageNet classification task, background masking improves performance by 0.5% (from 74.8% to 75.3%)  when applying it to just 2 large views and by 0.3% (from 76.8% to 77.1%) when applying it to the 4 large views and training with the 4 large views and 2 small views. While these performance gains are still significant, we would like to highlight that the saliency masking brings even more significant improvements on the robustness and out-of-distribution generalization downstream tasks. For this purpose, we conducted an additional ablation experiment, where we compare the performance of ReLICv2 without saliency masking (just using multi-crop) vs ReLICv2 with saliency masking. In the Table below, we report  Top-1 Accuracy under linear evaluation on ImageNetV2 and ImageNet-C (robustness), and ImageNet-R (IN-R), ImageNet-Sketch (IN-S) and ObjectNet (out-of-distribution). ImageNetv2 has three variants – matched frequency (MF), Threshold 0.7 (T-0.7) and Top Images (TI). The results for ImageNet-C (IN-C) are averaged across the 15 different corruptions. These results highlight that the use of saliency masking for ReLICv2 results in more robust representations.
>
> |  | MF | T-0.7 | Ti | IN-C | IN-R | IN-S | ObjectNet  |
> | -------- | --- | ----------- | --- | ----------- | --- | ----------- | --- |
> |     ReLICv2 w/o saliency masking      | 64.2%  |    73.1% |    78.0% |    43.8%     |       22.9%    |    5.7%   |     24.5% |
> | ReLICv2     | 65.3%  |    74.5% |  79.4%  |   44.8%      |    23.9% |      9.9%    |   25.9%
>
> We will include these ablation results and additional discussion about the benefits of saliency masking in the paper.

---

> ### Author Response · Authors · 2022-11-16
> **Follow-up**
>
> Dear reviewer Z39N,
>
> Thank you once again for your valuable feedback! We are sincerely grateful for your time and energy in the review process. We hope that our responses have addressed your concerns. If you have any remaining concerns, please let us know. We will do our utmost to address them.
>
> Thank you!
>
> Paper5023 Authors

---

> ### Author Response · Authors · 2022-12-06
> **Dear Reviewer Z39N**
>
> Thank you again for your time, commitment, constructive comments and interesting questions during the review process!
>
> We hope our responses (Nov 11) have addressed in detail the raised questions and comments. We have correspondingly updated the paper (Nov 18) and have furthermore released checkpoints and evaluation code for ReLICv2 (Nov 25).
>
> Please let us know if there are any outstanding questions or comments. We would be very happy to address these! :)
>
> If we have satisfactorily addressed the raised questions and concerns, we would kindly ask you to reevaluate your evaluation in light of this and consider raising your score.
>
> Thank you very much for your consideration of our paper!
>
> Paper5023 Authors

---

### Official Review · Reviewer_jUzd · 2022-10-25

**Confidence:** 3
**Correctness:** 3
**Technical Novelty And Significance:** 3
**Empirical Novelty And Significance:** 4
**Recommendation:** 6

**Clarity, Quality, Novelty And Reproducibility:**

Clarity: This is a well written paper. It is very clear.

Quality and Novelty: The tricks proposed in this paper is very effective as shown in the results. These tricks seem to be general purpose and would benefit other papers too.

Reproducibility: The authors have provided extensive experiment details in the appendix which is a good sign for reproducibility.

**Strength And Weaknesses:**

Strengths:

This is a well written paper. The authors clearly state what their main contribution is. The two tricks that are proposed is intuitive and looks promising. The paper is also explained well and I was able to understand the main contributions.

The experiments are very extensively performed. Comparisons with existing Resnet based models are included. Experiments are performed with different model sizes. Results are also shown on OOD, transfer, semi-supervised leanrning and for other tasks such as segmentation. Experimental section is quite strong and I don't have any concerns.

I also apprecitate the authors for providing detailed experimental settings in appendix.

Weaknesses:

The main tricks proposed in this work seem to be general purpose. Do you think this would work on ViTs?
In this regard, the self-attentions in ViTs might alrealy be doing some sort of saliency detection. For instance, they have been shown to do semantic segmentation. So, I am curious if thesr tricks can be used with ViTs?

Why did you stick only with Resnet based architectures? Do you think ReLICv2 would work with architectures like Swin? If so, would you set SOTA results there?




**Summary Of The Paper:**

This paper proposes a few tricks that can be added on top of ReLIC for self-supervised learning. First, saliency masking is used to explicitly reduce spurious correlations in the model. Second, augmentations of small sizes are used. This has the effect of occluding some regions of the image. Using these two tricks, the authors were able to improve the performance of ReLIC by a good margin. Moreover, the authors also achieve SOTA results on Resnet based self-supervised learning methods. Several ablation studies are also performed.



**Summary Of The Review:**

Overall, I think this is a very good paper in terms of ideas proposed and rigorous experimental study. I advocate for acceptance.


======================================

Post rebuttal:

After looking at other reviews and having discussion with some reviewers, I feel like the paper has the following issues:
Similarity with prior work: As pointed out by reviewer Z39N, background augmentation only has marginal improvements. So, this makes me a bit worried about the main message the paper gives. Its probably that improvements mainly comes because of multi-crop at scale. If this is the case, the authors need to make this very clear (if the paper gets accepted). I also agree with other reviewers that claiming that they beat supervised learning is bit of a overclaim and I would appreciate if the authjors address this.

But, I also feel that even if the paper proposes a few tricks, I am fine with accepting the paper as the computer vision field has made progress through a series of tricks. But the authors should make that message very clear.

Since I was not aware of these concerns at the time of giving my review, I downvote my rating to 6.

---

> ### Author Response · Authors · 2022-11-11
> **Reply for reviewer jUzd**
>
> We thank the reviewer for their time and effort in reviewing our paper and for their insightful comments and suggestions on how to improve the paper.
>
> Please allow us to clarify below the reviewer’s concerns.
>
> The inductive biases proposed in this paper (saliency masking and multiple crops of varying sizes) are indeed general purpose. Thank you for highlighting that!
>
> These inductive biases can be applied both as part of other self-supervised methods (e.g. SimCLR, BYOL) and with other architectures such as ViTs as they are part of the data input pipeline. We focussed on combining these biases with the strongly performing ReLIC method and used ResNets of widely varying sizes to illustrate the performance and generality of our method. We focussed on ResNets as they are commonly used in computer vision and are also at the moment more extensively used within the self-supervised literature than ViTs. The recent developments in ViTs model are very promising and we believe that our proposed inductive biases together with invariance will enable ViTs to learn more informative representations. The general trends we observe in Figure 6 give us reason for optimism; ReLICv2 outperforms recent self-supervised methods (e.g. DINO, MoCov3) and has similar performance to more powerful architectures with more involved training procedures (e.g. EsViT) for a comparable number of parameters. We are starting to explore the combination of ReLICv2 and ViTs, and will hopefully be able to empirically validate improved performance also with ViTs very soon.

---

> ### Author Response · Authors · 2022-11-16
> **Follow-up**
>
> Dear reviewer jUzd,
>
> Thank you once again for your valuable feedback! We are sincerely grateful for your time and energy in the review process. We hope that our responses have addressed your concerns. If you have any remaining concerns, please let us know. We will do our utmost to address them.
>
> Thank you!
>
> Paper5023 Authors

---

> ### Author Response · Authors · 2022-12-12
> **Response to post rebuttal comments**
>
> Dear Reviewer,
>
> Thank you for your additional comments! Please allow us to address them below.
>
> ### Improvements from background masking
> Please note that, as we have also mentioned in the response to Reviewer Z39N, we already perform ablations in the paper where we show that for evaluation on the ImageNet classification task, background masking improves performance by 0.5% (from 74.8% to 75.3%)  when applying it to just 2 large views and by 0.3% (from 76.8% to 77.1%) when applying it to the 4 large views and training with the 4 large views and 2 small views (please see section 3.1 for details).
>
> Nevertheless, saliency masking brings important improvements on the robustness and out-of-distribution generalization downstream tasks. In the additional ablation experiment we have conducted during the rebuttal period, we demonstrate this by comparing the performance of  ReLICv2 without saliency masking (just using multi-crop) vs ReLICv2 with saliency masking. In the Table below (please note that we have already included these results in Appendix G.3 in the revised version of the paper), we report  Top-1 Accuracy under linear evaluation on ImageNetV2 and ImageNet-C (robustness), and ImageNet-R (IN-R), ImageNet-Sketch (IN-S) and ObjectNet (out-of-distribution). ImageNetv2 has three variants – matched frequency (MF), Threshold 0.7 (T-0.7) and Top Images (TI). The results for ImageNet-C (IN-C) are averaged across the 15 different corruptions. These results highlight that the use of saliency masking for ReLICv2 results in more robust representations.
>
> |  | MF | T-0.7 | Ti | IN-C | IN-R | IN-S | ObjectNet  |
> | -------- | --- | ----------- | --- | ----------- | --- | ----------- | --- |
> |     ReLICv2 w/o saliency masking      | 64.2%  |    73.1% |    78.0% |    43.8%     |       22.9%    |    5.7%   |     24.5% |
> | ReLICv2     | 65.3%  |    74.5% |  79.4%  |   44.8%      |    23.9% |      9.9%    |   25.9%
>
>
>
> ### Key message of the paper
> The key message of the paper is in showing that enforcing invariance over two simple and general inductive biases can significantly improve self-supervised representations so that they achieve state-of-the-art performance across a very wide range of diverse datasets, architectures and tasks, as well as pre-training datasets.
>
> ### Supervised baseline
> In our work we also examine a like-for-like comparison of self-supervised and supervised learning. To achieve this we use the same architectures and data augmentations for both the supervised and ReLICv2 models in order to ensure a fair comparison. As the supervised baseline, we use 76.5% as it is used as the supervised baseline in the SSL literature. Note that this is the supervised performance under the augmentations introduced in the SimCLR paper (Note that we only add saliency masking on top of the SimCLR augmentations and it has been shown in the literature that saliency masking has a detrimental effect on supervised performance (we also discuss this in section 3)).  Supervised performances above this number are predominantly achieved by using additional complex and sophisticated augmentation strategies and a number of intricate optimization tricks like casting parameters between different levels of precision. At the beginning of section 3 we comment on the existence of improved supervised models based on improved augmentation and optimization approaches. We believe that these additional tricks can be easily applied within self-supervised methods and we expect them to lead to analogous improvements as in supervised learning.
>
> We understand how the statement of beating supervised learning can be viewed as an overclaim. We will further clarify the main contributions of the paper in the manuscript and add more context around the comparison to supervised learning upon acceptance.

---

### Official Review · Reviewer_aWZZ · 2022-11-02

**Confidence:** 4
**Correctness:** 3
**Technical Novelty And Significance:** 2
**Empirical Novelty And Significance:** 2
**Recommendation:** 5

**Clarity, Quality, Novelty And Reproducibility:**

=================================================================

[Technical Comments]

1) Does the linear evaluation setup in fact disclose learning less spurious/correlated features? Background and style features may not be useful for dominant/foreground object classifications, but they would be valuable for dense prediction, which often needs consistency of multiple objects.
2) Furthermore, I believe the linear evaluations employ all the labels in the fine-tuning stage, thus the performances shown in the paper are not label-free as the title implies. On the other hand, studying the labeled sample complexity will have more practical implications and will be a more engaging story.
3) In Tables 4 and 5, how does the ReLICv1 perform?

=================================================================



**Strength And Weaknesses:**

=================================================================

[Main Strengths]

The key merit of this paper is that the authors' approaches for improving the ReLIC indeed produce better benchmark results than previous ResNet-based methods at a different model and dataset sizes.

=================================================================

[Main Weaknesses]

The fundamental shortcoming of this study is that, while the basic idea is interesting, it does not appear to have the intended impact, as the title implies, much like the various flavors of contrastive methods. Given the richness of learned visual representation, focusing primarily on linear evaluations in the classification tasks is a too narrowed contribution.  I encourage authors to explore further into specific image domains and/or paired with Vision transformers to see whether ReLICv2 is still effective. Furthermore, I don’t see the implementation GitHub repo is intended to be provided.

=================================================================


**Summary Of The Paper:**

=================================================================

[Summary]

This paper proposes an improved version of ReLIC, dubbed ReLICv2, by including views of varying sizes and saliency masking into the ReLIC's training loss, and presents benchmarking results using ResNet encoders in several experiments.

=================================================================



**Summary Of The Review:**

Using views of varying sizes and saliency masking to improve the original ReLIC's training loss is intriguing, but, it does not appear to make a significant impact, as do the many flavors of contrastive self-supervised approaches.

---

> ### Author Response · Authors · 2022-11-11
> **Reply for reviewer aWZZ [Part 2/2]**
>
> ### Pairing ReLICv2 with Vision transformers
>
> Please note that the inductive biases proposed in this paper (saliency masking and multiple crops of varying sizes) are general purpose. Thus, they can be applied both as part of other self-supervised methods (e.g. SimCLR, BYOL) as well as in conjunction with other architectures such as ViTs as they are part of the data input pipeline. In our work we focussed on combining these inductive biases with the strongly performing ReLIC loss and used ResNet architectures of widely varying sizes to showcase the performance and generality of our proposed approach. We focussed on Resnet architectures as they are commonly used across many computer vision tasks and are also at the moment more extensively used within the self-supervised literature than ViTs. We find the recent developments in ViTs exciting and we believe that these inductive biases as well as invariance will enable ViTs to learn more informative representations. The general trends we observe in Figure 6 give us reason for optimism; ReLICv2 outperforms recent self-supervised methods (DINO, MoCov3) and has similar performance to more powerful architectures with more involved training procedures (EsViT) for a comparable number of parameters. We are starting to explore the combination of ReLICv2 and ViTs, and will hopefully be able to empirically validate improved performance also with ViTs very soon.
>
> ### Implementation
>
> We are currently working hard to make publicly available the evaluation code and checkpoints for the pre-trained ResNet models using ReLICv2 as soon as possible.
>
> ### Linear evaluation set-up disclosing less spurious features + dense prediction tasks (Technical comment 1)
>
> Please note that during ReLICv2 pre-training, the background masking is only applied with a small probability $p_m = 0.1$ as part of the data augmentations pipeline. Thus, the learnt representations are not completely disregarding the background, but instead are more robust to spurious correlations that often happen between the objects in the image and the background. We believe that the linear evaluation set-up on the OOD and generalization downstream tasks does indeed show that ReLICv2 learns representations that are less prone to spurious correlations over competing methods. Moreover, we have conducted additional ablation experiments to show the importance of saliency masking on downstream performance, particularly on robustness and OOD generalization downstream tasks. In the table below, we compare the performance of ReLICv2 without saliency masking (just using multi-crop) vs ReLICv2 with saliency masking. We report  top-1 accuracy under linear evaluation on ImageNetV2 and ImageNet-C (robustness), and ImageNet-R (IN-R), ImageNet-Sketch (IN-S) and ObjectNet (out-of-distribution). ImageNetv2 has three variants – matched frequency (MF), Threshold 0.7 (T-0.7) and Top Images (TI). The results for ImageNet-C (IN-C) are averaged across the 15 different corruptions. These results highlight that the use of saliency masking for ReLICv2 results in more robust representations.
>
> |  | MF | T-0.7 | Ti | IN-C | IN-R | IN-S | ObjectNet  |
> | -------- | --- | ----------- | --- | ----------- | --- | ----------- | --- |
> |     ReLICv2 w/o saliency masking      | 64.2%  |    73.1% |    78.0% |    43.8%     |       22.9%    |    5.7%   |     24.5% |
> | ReLICv2     | 65.3%  |    74.5% |  79.4%  |   44.8%      |    23.9% |      9.9%    |   25.9%
>
>
> **Evaluation on dense prediction task**: Moreover, in Section 4.2 we show that we can successfully transfer the representations learnt by ReLICv2 to PASCAL (Everingham et al., 2010) and Cityscapes (Cordts et al., 2016) semantic segmentation tasks, which represent dense prediction tasks. In particular ReLICv2 significantly outperforms BYOL on both datasets and on PASCAL and ReLICv2 also outperforms DetCon (Henaff et al., 2021), a method specifically trained for detection on PASCAL. This further highlights that the inductive biases used by ReLICv2 (multiple crops of varying sizes and using saliency masking with a small probability during training) result in strong performance of a wide range of vision tasks.
>
> ### Linear evaluation employs labels (Technical comment 2)
>
> ReLICv2 does not utilize labels in any way for learning representations.
> For linear evaluation, we follow the standard protocol in the literature (e.g. in SimCLR, BYOL, SwaV etc.) which uses labels just to train a single linear layer on top of the frozen, pretrained representation. Thus, labels are just used to evaluate the quality of the representation, but not to learn the representation itself.
>
> ### Performance of ReLICv1 in Tables 4 and 5 (Technical comment 3)
>
> We are currently running experiments to provide the reviewer with the performance of ReLICv1 on the semantic segmentation downstream task and in the setting of pre-training it on large and uncurated datasets such as JFT.

---

> ### Author Response · Authors · 2022-11-11
> **Reply for reviewer aWZZ [Part 1/2]**
>
> We thank the reviewer for their time and effort in reviewing our paper and for their insightful comments and suggestions on how to improve the paper.
>
> Please allow us to clarify below the reviewer’s concerns.
>
> ### Evaluating the richness of the learned visual representations
>
> While we indeed focus on linear evaluation in the classification tasks on ImageNet to show (for the first time in the literature) that it is possible to achieve better performance than supervised models across a wide range of ResNet architectures, we also evaluate the representations learnt by ReLICv2 on several image domains and downstream tasks that require fine-tuning of the learnt representations.
> In particular, we:
> - (Section 4.2) Evaluate transfer performance of the representations on a wide range of classification benchmarks under both the linear and fine-tuning protocols (details on these are given in the appendix). We show that the representations learnt by ReLICv2 improve upon both the supervised baseline and competing methods.
> - (Section 4.2) Evaluate performance on fine-tuning the representations on the semantic segmentation downstream task and we show that on the PASCAL dataset (Everingham et al., 2010), RELICv2 outperforms DetCon (Henaff et al., 2021), a method specifically trained for detection.
> - (Appendix B.2) Evaluate performance in a semi-supervised learning setting where we finetune both the pre-trained encoder and a randomly initialized linear classifier with either 1% or 10% of the ImageNet training data. In this setting ReLICvs outperforms both the standard supervised baseline and all previous state-of-the-art self-supervised methods when using 10% of the data for fine-tuning, and archives competitive performance with C-BYOL in the 1% fine-tuning regime.
>
> In addition, please note that linear evaluation is the standard evaluation methods used in the very extensive self-supervised learning literature to demonstrate the performance of the learnt representations on downstream tasks (Chen et al., 2020a, Grill et al., 2020, Kuang-Huei Lee et al., 2021 and many more) and this enables us to perform a meaningful evaluation against competing methods. Moreover, compared to related works, we perform a more extensive evaluation of the learnt representations and demonstrate state-of-the-art performance on the following tasks (in addition to the transfer and semi-supervised learning tasks mentioned above):
> - (Section 3) Linear evaluation on the ImageNet test set for ResNet50 1x, ResNet50 2x, ResNet50 4x, ResNet101, ResNet152, ResNet200, ResNet 200 2x encoder architectures. - where we show that is it possible to learn representations in a self-supervised manner, without using any labeled information, that achieve better performance than representations learnt by supervised learning methods
> - (Section 4.1) Robustness (ImageNetV2 and ImageNet-C datasets) and out-of-distribution generalization (ImageNet-R, ImageNet-Sketch, ObjectNet datasets) downstream tasks.
> - (Section 4.3 and Appendix D) Pretraining representation on large and curated datasets such as the Joint Foto Tree (JFT-300M) dataset which consists of 300 million images from more than 18k classes. The representations learnt by pre-training on JFT-300M are again evaluated on a wide-range of downstream tasks including linear-evaluation on the ImageNet dataset, transfer learning (fine-tuning) on 11 different datasets, and robustness and OOD generalization downstream tasks.
>
> Overall, we believe that the wide range of downstream tasks, settings and pretraining datasets used for evaluation demonstrate the generality and wide-range applicability of the representations learnt by ReLICv2, thus representing an important contribution to the self-supervised learning community.

---

> ### Author Response · Authors · 2022-11-16
> **Follow-up**
>
> Dear reviewer aWZZ,
>
> Thank you once again for your valuable feedback! We are sincerely grateful for your time and energy in the review process. We hope that our responses have addressed your concerns. If you have any remaining concerns, please let us know. We will do our utmost to address them.
>
> Thank you!
>
> Paper5023 Authors

---

> ### Author Response · Authors · 2022-11-18
> **Additional results**
>
> We thank the reviewer again for their time and thoughtful comments which have helped us to improve our paper. To address the Technical Comment 3 raised by the reviewer, we have now conducted additional experiments to evaluate ReLIC [Mitrovic et al., 2021] on semantic segmentation and have pre-trained ReLIC on the large and uncurated dataset JFT-300M as suggested by the reviewer.
> - In Table 4, on the semantic segmentation downstream tasks, we have now added results for ReLIC. ReLIC achieves 77.7 mIoU on PASCAL and 75.0 mIoU on CityScapes. We note that ReLICv2 still provides a noticeable improvement over ReLIC in the semantic segmentation tasks.
> - In Table 5, for pretraining on the JFT-300M dataset and evaluating on ImageNet, for the ReLIC method we get results reported in the below table. Note that during this discussion period we did not have the compute resources needed to run ReLIC for 5000 epochs. From the table we see a significant improvement in the learned representation between ReLIC and ReLICv2 on the large-scale, uncurated dataset JFT. We also see that there is no performance improvement when training ReLIC on JFT for longer than 1000 epochs.
>
> | Method | 1000 epochs  | 3000 epochs | $>$4500 epochs |
> | -------- | ---- | ----------- | ----------- |
> | ReLIC  | 61.5 | 61.5 | -- |
> | BYOL    |  67.0 |  67.6 | 67.9 (5000 epochs) |
> | Divide and Contrast  | 67.9 | 69.8 | 70.7 (4500 epochs) |
> | ReLICv2 (ours) |  70.3 | 71.1 |  71.4 (5000 epochs) |

---

> ### Author Response · Authors · 2022-12-06
> **Dear Reviewer aWZZ**
>
> Thank you again for your time, commitment, constructive comments and interesting questions during the review process!
>
> We hope our responses (Nov 11 & Nov 18) have addressed in detail the raised questions and comments. We have correspondingly updated the paper (Nov 18) and have furthermore released checkpoints and evaluation code for ReLICv2 (Nov 25).
>
> Please let us know if there are any outstanding questions or comments. We would be very happy to address these! :)
>
> If we have satisfactorily addressed the raised questions and concerns, we would kindly ask you to reevaluate your evaluation in light of this and consider raising your score.
>
> Thank you very much for your consideration of our paper!
>
> Paper5023 Authors

---

### Public Comment · ~Chaitanya_Ryali1 · 2022-11-17
**Discussion regarding large overlap with our prior work**

Dear authors,

Thanks for citing our prior work [1] as having previously used background augmentations (BG Augs). Given the large overlap with our prior work (detailed below), we feel a more detailed discussed in your work is warranted.

Some examples of core contributions of your work that overlap with our prior work:

1) Adapting DeepUSPS to produce an unsupervised saliency model and using it for BG Aug
2) Showing BG Aug can benefit multiple SSL methods
3) Characterizing the impact of mask quality and studying mask corruptions
4) Showing BG Aug can improve multiple downstream tasks, especially in distribution-shift settings and improved label efficiency

Beyond these, our work investigated many variants of BG Augs across SSL methods (MoCov2, BYOL, SwAV, DINO), training durations, architectures (CNNs, ViTs), saliency methods and in numerous downstream and distribution shift-settings. We also studied when and how BG Augs can provide benefits. Of particular note,

1) we were able to achieve performance *on-par* with the standard supervised baseline on ImageNet, when BG Aug is used together with SwAV,
2) though self-supervised ViTs (e.g. DINO) are known to be particularly good at separating foreground and background, we’ve shown [2] *even ViTs*  benefit from a similarly large boost in performance by using BG Augs

Thanks,

Chay, David, Ari


1. Characterizing and Improving the Robustness of Self-Supervised Learning through Background Augmentations, Ryali, Schwab, Morcos, arxiv, 2021
2. Learning Background Invariance Improves Generalization and Robustness in Self Supervised Learning on ImageNet and Beyond,  Ryali, Schwab, Morcos, ImageNet PPF Workshop, NeurIPS 2021,  https://slideslive.com/38974549

---

> ### Author Response · Authors · 2022-12-06
> **Refer to main thread**
>
> Please refer to the main thread for a detailed response to your comment.

---

### Author Response · Authors · 2022-11-18
**Updated version of the paper**

We have uploaded a revised version of the paper which includes the following changes suggested by the reviewers. The changes are indicated in blue in the updated manuscript.
- In Appendix F.2.2 we have added examples of the saliency masks obtained from the saliency detection network and a discussion of failure cases.
- In Appendix G.3 we have included the additional ablation experiments to show the importance of saliency masking on robustness and OOD generalization downstream tasks.
- In Table 4, on semantic segmentation tasks, we have now added results for ReLIC. ReLIC achieves 77.7 mIoU on PASCAL and 75.0 mIoU on CityScapes. We note that ReLICv2 still provides a noticeable improvement over ReLIC on semantic segmentation tasks.
- In Table 5,  for pretraining on the JFT-300M dataset and evaluating on ImageNet, we have included the results for pre-training ReLIC for 1000 and 3000 epochs. Note that during this discussion period we did not have the compute resources needed to run ReLIC for 5000 epochs. From the table we see a significant improvement in the learned representation of ReLICv2 over ReLIC on the large-scale, uncurated dataset JFT.
- We have expanded the discussion in the Related Work section (Section 5) to provide a more thorough comparison with [1, 2].

[1] Characterizing and Improving the Robustness of Self-Supervised Learning through Background Augmentations, Ryali, Schwab, Morcos, arxiv, 2021

[2] Learning Background Invariance Improves Generalization and Robustness in Self Supervised Learning on ImageNet and Beyond, Ryali, Schwab, Morcos, ImageNet PPF Workshop, NeurIPS 2021, https://slideslive.com/38974549

---

> ### Public Comment · ~Chaitanya_Ryali1 · 2022-11-19
> **we believe our work is prior, not concurrent**
>
> Dear authors,
>
> Thanks for your response and the updated discussion. Given our initial manuscript was posted in march 2021 [1], and our update in sept 2021 [2], which even more overlaps with your work, we believe our work should be considered *prior* work rather than concurrent (as you currently discuss it). Moreover, as we previously noted, our work also demonstrates robustness and OOD benefits as your work does and has also investigated ViTs and not just ResNet-50 as you currently discuss.
>
> 1. https://arxiv.org/abs/2103.12719
> 2. https://openreview.net/forum?id=zZnOG9ehfoO
>
> Thanks,
>
> Chay, David, Ari

---

> > ### Author Response · Authors · 2022-12-06
> > **Refer to main thread**
> >
> > Please refer to the main thread for a detailed response to your comment.

---

### Author Response · Authors · 2022-11-19
**Response to comment about overlap with prior work**

Thank you for your comment! Please allow us to address your concerns below.

As pointed out, we cite the relevant work [1] in our paper ([2] appears to be a workshop version of [1] and we have added this citation). We agree that there are several similarities between our paper and [1, 2], which we already discuss in our submission and outline more context below, a summary of which we have included in our manuscript. The significant differences between our methods that can be divided into these two broad categories:

- Methodological differences
- Differences in experimental results


**Methodological differences:**

- The main contribution of our paper is to extend the self-supervised method ReLIC [4] with two additional inductive biases to avoid learning spurious correlations and obtain more informative representations. Note that [1,2] do not explore the use of background augmentations with the ReLIC loss (although ReLIC [4] came out 5 months before the first version of [1]) and also do not explore the importance of multiple views of varying sizes for learning informative representations. This is in contrast to our work which shows that a large improvement in representation quality can be achieved through using multiple views. In particular, we find that the multiple data views of varying sizes improve performance by 2% over ReLIC (from 74.8% to 76.8%), while saliency masking adds another +0.3% on that.

- The focus of our paper is to explore whether it is possible to achieve **better** than supervised performance on ResNet architectures, rather than to mainly investigate the benefits of background augmentations when combined with a large number of self-supervised methods on the ResNet 50 1x encoder as it is for [1, 2]. Note also that we do not claim novelty over proposing to use background removal in self-supervised learning as this was already proposed in prior work [3] (which is also cited in [1, 2]). **Our contribution is in showing that two simple and general inductive biases can significantly improve self-supervised representations so that they outperform supervised baselines on a wide range of ResNet architectures (ResNet 50 1x, ResNet 50 2x, ResNet 50 4x, ResNet 101, ResNet 152, ResNet 200, ResNet 200 2x).**

- Note that we train the saliency detection network directly on a subset of images from ImageNet rather than on the additional dataset MSRA-B as it is done in [1, 2]. This means that our self-supervised learning pipeline uses no extra data, thus ensuring a fair comparison between competing methods that also only use the ImageNet training set. Furthermore, this allows us to make a fair comparison between the performance of our method ReLICv2 and the supervised baseline. Note that we also do not use the same set of hand-crafted methods as [1, 2].


**Experimental differences:**

Unlike [1, 2] which focus on the ResNet50 1x architecture, in our paper we demonstrate state-of-the art performance of ReLICv2 for a wide range of ResNet encoder architectures spanning different depths and widths and parameter counts (from 24 to 375 million parameters) -- ResNet50 1x, ResNet50 2x, ResNet50 4x, ResNet101, ResNet152, ResNet200, ResNet 200 2x. Crucially, this is the first work that demonstrates better than supervised performance on such a wide range of encoders. Note that in [1, 2] the authors show improvement in combining background augmentations with some self-supervised methods, but achieve much lower performance and only focus on a single ResNet architecture. (The best performance reported in [1] for SWaV with ResNet 50 1x encoder + background removal is 76.1%, while ReLICv2 achieves 77.1% for the same encoder architecture (supervised model achieves 76.5%).)

Similarly to [1, 2] we also find that background augmentations bring significant improvements for robustness and out-of-distribution generalization downstream tasks. A promising direction for future work would be to explore this in conjunction with multiple data views of varying sizes and the explicit invariance loss in order to achieve even better performance.

In the first version of [1] (March 2021), a supervised saliency method was used and thus the resulting learned representations are not self-supervised; we focus on an unsupervised saliency masking pipeline and thus ensure that our representations are self-supervised.
In the second version of [1] (November 2021) and workshop submission [2] (end of September 2021), an unsupervised saliency masking method was used. We would also like to note that the first version of work is concurrent to the second version of [1].

---

> ### Author Response · Authors · 2022-11-19
> **Regarding Vision Transformers**
>
> Regarding the below comment on Vision Transformers (ViTs), we searched the 67 page document of [1] and only found one mention of  Vision Transformers in the text (outside of references in which there are 3 mentions). The mention in the text (in the last section in the last paragraph before the Acknowledgements) reads:
>
> “There has been increasing recent interest (Chen et al., 2021; Caron et al., 2021; Li et al., 2021a) in self-supervised learning for Vision Transformers (ViTs, Dosovitskiy et al. 32 Background Augmentations for Self-Supervised Learning (2021)). Future work could investigate whether background (or foreground) augmentations can benefit SSL for ViTs.”
>
> As such, we are genuinely puzzled where and how ViTs are investigated as claimed in the comment below.
>
> [1] Characterizing and Improving the Robustness of Self-Supervised Learning through Background Augmentations, Ryali, Schwab, Morcos, arxiv, 2021
>
> [2] Learning Background Invariance Improves Generalization and Robustness in Self Supervised Learning on ImageNet and Beyond, Ryali, Schwab, Morcos, ImageNet PPF Workshop, NeurIPS 2021, https://slideslive.com/38974549
>
> [3] Zhao, Nanxuan, et al. "Distilling localization for self-supervised representation learning." Proceedings of the AAAI Conference on Artificial Intelligence. Vol. 35. No. 12. 2021.
>
> [4] Mitrovic, Jovana, et al. "Representation learning via invariant causal mechanisms." International Conference on Learning Representations (2021).

---

### Author Response · Authors · 2022-11-25
**Opensourcing ReLICv2**

Dear reviewers,

We have now also released the evaluation code and checkpoints for ReLICv2 for all of the ResNet encoder architectures used in our paper (7 encoders in total - ResNet 50 1x, 2x, 4x, ResNet 101, ResNet 152, ResNet200 1x, 2x).

The code can be found here: https://drive.google.com/drive/folders/1_Ak1OUV-vzT957oKzlW8UENTyptOAFWD?usp=share_link

Please refer to the README.md for details on how to install the required packages, run the evaluation code and access the checkpoints.

We hope this addresses the reviewers’ concerns regarding the availability of the code. Moreover, we hope that the released  ReLICv2 pre-trained checkpoints (and corresponding evaluation code) will be a valuable contribution to the machine learning community.

---

### Author Response · Authors · 2022-12-13
**Thank you**

Dear Reviewers, ACs, SACs and PCs,

Thank you very much for your time and effort in considering our work.
Thank you also for the constructive comments, questions and thoughtful discussion.

Best wishes,
Paper5023 Authors

---

### Decision · Program_Chairs · 2023-01-20

**Decision:**

Reject

**Justification For Why Not Higher Score:**

There is no strong champion of the paper. The area chairs found that the weaknesses raised by the reviewers are valid and weighted the paper's weaknesses over the current strengths. Therefore, the area chairs cannot recommend the acceptance of the current paper. But the area chairs believe the paper will be significantly improved by (i) revisiting several statements such as beating supervised learning and the precise contribution of background masking, and (2) including additional experiments such as comparing with up-to-date supervised baseline implementation and using Vision Transformers.

**Justification For Why Not Lower Score:**

N/A.

**Metareview: Summary, Strengths And Weaknesses:**

**Summary**: The authors propose a self-supervised representation learning approach RELICv2, which builds upon the previous self-supervised method RELIC [Mitrovic et al., 2021]. There are two key additions: (i) unsupervised saliency masking that decouples the foreground from background, and (ii) multiple data augmentation of varying sizes especially with small sizes. RELICv2 is claimed to be the first unsupervised representation learning approach that consistently outperforms the supervised learning baseline across a range of ResNet architectures and various learning scenarios (ImageNet evaluation, transfer, robustness, and OOD generalization)..

**Strengths And Weaknesses**: From the reviewers’ perspectives, the strengths of the paper are clear: simple strategies, strong and state-of-the-art performance, and extensive evaluation.

On the other hand, there are several concerns raised by the reviewers.

(i) Over-claiming. The reviewers felt that the main statement that RELICv2 for the first time beats supervised learning is a bit over-claiming. In particular, the reviewers pointed out that the supervised learning baseline compared in this paper comes from Pytorch's official implementation some time ago. Recently, the supervised performance has been significantly increased with better augmentation and training recipes. Therefore, a fair comparison with the up-to-date supervised baseline is needed to fully support the claim in the paper.

(ii) Limited technical novelty. While the performance of RELICv2 is strong, all reviewers felt that the key techniques of background masking and multiple crops were already proposed in the previous work.

(iii) Marginal performance contribution from background masking. The reviewers pointed out that while background masking is stated as one of the key techniques, the ablation study shows that it only brings in marginal improvements; the performance improvements mainly come from multi-crop augmentation at scale. The authors’ response confirmed this on the ImageNet classification task, but mentioned that the improvements from background masking become significant on the robustness and OOD generalization downstream tasks. The reviewers were concerned about the imprecise statement in the paper.

(iv) The paper claims that the two strategies are general purpose. The reviewers were interested in the performance when the strategies are paired with Vision Transformers. These results were not provided in the response due to time constraint.

At the end of the discussion phase including the AC-review meeting, there is a lack of a strong champion of the paper. The area chairs found that the weaknesses raised by the reviewers are valid and weighted the paper's weaknesses over the current strengths. Therefore, the area chairs cannot recommend the acceptance of the current paper. But the area chairs believe the paper could be significantly improved by (i) revisiting several statements such as beating supervised learning and the precise contribution of background masking, and (2) including additional experiments such as comparing with up-to-date supervised baseline implementation and using Vision Transformers.

**Summary Of Ac-Reviewer Meeting:**

In the AC-reviewer meeting, reviewers and area chairs discussed the key strengths and weaknesses summarized above.

Reviewer jUzd recognized the limited novelty, meanwhile favored small incremental steps that show big performance improvement. That's the main reason Reviewer jUzd liked the paper initially and gave a score of 8.

Reviewer Z39N pointed out that the paper combines a few existing effective building components including the RELIC loss function, multi-crops, and background masking. Even though these components are already existing in the literature, Reviewer Z39N found that some special settings including 4x big + 2x small multi-crops are novel. The main concern from Reviewer Z39N is that the ablation study shows that the background masking only yields marginal improvements. In addition, Reviewer Z39N pointed out the supervised baseline compared in the paper is somewhat outdated, making the key statement of the paper an overclaim.

After the discussion among the reviewers, Reviewer jUzd agreed that the concerns are valid, and thus decreased the score to 6.

In addition, Reviewer Z39N pointed out the overlap with prior/concurrent work of [Ryali et al., 2021]. There is also public discussion on the OpenReview system, and authors clarified the difference between the two papers. Reviewer Z39N felt that the key ideas are very similar with different instantiations.

At the end of the AC-review meeting, there is no strong champion of the paper. The area chairs found that the weaknesses raised by the reviewers are valid and weighted the paper's weaknesses over the current strengths. Therefore, the area chairs cannot recommend the acceptance of the current paper. But the area chairs believe the paper will be significantly improved by (i) revisiting several statements such as beating supervised learning and the precise contribution of background masking, and (2) including additional experiments such as comparing with up-to-date supervised baseline implementation and using Vision Transformers.